# Effects of nomadic grazing system and indoor concentrate feeding systems on performance, behavior, blood parameters, and meat quality of finishing lambs

Imaneh Sadrarhami[1]*, Masoud Alikhani[1], Ebrahim Ghasemi[1], Amir Hossein Mahdavi[1], Nafiseh Soltanizadeh[2], Maria Font-i-Furnols[3], Morteza Hosseini Ghaffari[4]*

1 Department of Animal Sciences and Technology, College of Agriculture, Isfahan University of Technology, Isfahan, Iran, 2 Department of Food Science, College of Agriculture, Isfahan University of Technology, Isfahan, Iran, 3 IRTA-Food Industries, Finca Camps i Armet, Monells, Spain, 4 Institute of Animal Science, University of Bonn, Bonn, Germany

* morteza1@uni-bonn.de (MHG); sadrearhamiimaneh@yahoo.com (IS)

## Abstract

The objective of the study was to evaluate the effects of three production systems on growth performance, behavior, blood parameters, carcass characteristics, and meat quality. A total of 30 lambs (n = 10 lambs/treatment) were randomly assigned to three production systems that included rotational grazing (NG) and two different levels of concentrate (CON), one with medium (roughage/concentrate ratio 50:50% based on DM, MC) and one with high concentrate (roughage/concentrate ratio 30:70% based on DM, HC) during the 90-day fattening period. At the start of the experiment, all lambs averaged 90 ± 4 days of age (mean ± SD) and were slaughtered at an average of 180 ± 3 days (mean ± SD). CON-fed lambs had higher average daily gain and loin thickness than NG-fed lambs. The NG lambs spent more time eating, drinking, and standing, but less time resting and rumination than the CON-fed lambs. In addition, plasma lipid, β-hydroxybutyrate, and urea levels were higher, while phosphorus levels were lower in NG-fed lambs than in CON-fed lambs. CON-fed lambs had better carcass yield, but gastrointestinal tract and rumen weights were lower than NG lambs. CON-fed lambs had higher pH values 0 h *post mortem* than the NG lambs; however, there was no effect of treatment on pH 24 h *post mortem*. The *post-mortem* color of the LD muscle of NG lambs had a higher lightness and yellowness index and a lower redness index than that of the LD muscle of CON-fed lambs. The results of this study showed that lambs fed CON had better carcass yield than lambs fed NG, although feed intake, feed conversion ratio (FCR), growth performance, carcass yield, and meat quality of lambs fed MC and HC were similar.

## Introduction

Consumers are increasingly interested in sustainable and animal-friendly production and healthy diets [1]. Meat from lambs raised on pasture contains less subcutaneous backfat than

**Data Availability Statement:** All relevant data are within the paper and its Supporting Information files.

**Funding:** The author(s) received no specific funding for this work.

**Competing interests:** The authors have declared that no competing interests exist.

meat from lambs raised in pens, which is more attractive to consumers [2]. Diaz et al. [2] reported that in sheepfold lamb dorsal fat thickness, kidney knob and cannel fat, and percent leg fat were higher in lambs raised in pens than in lambs raised on pasture. Lambs with higher weight had more fat. Considering that feed costs often represent a significant portion of the total variable costs of lamb production, this favorable environment for abundant grass growth results in pasture grass being the most economical feed source. In addition, pasture grass has a high content of natural antioxidants and therefore provides animals with better protection against lipid oxidation [3]. According to Santé-Lhoutellier et al. [4], the oxidative stability of lamb is related to its diet. Pasture feeding offers some advantages over concentrate feeding in terms of lipid oxidation and, to a lesser extent, protein oxidation [4].

There has been much discussion about raising grass-fed lambs because lambs fed concentrates are more efficient than lambs raised on pasture [5]. By giving concentrates to lambs in the barn, it is possible to improve weight gain, carcass yield, and ultimately the profitability of the production system [6]. Color of meat is also an important factor in consumer selection. In particular, exercise can influence color and flavor [7]. It has been shown that cows that are constantly on pasture or kept in an extensive environment have a darker muscle color [8].

Therefore, the appropriate production system and the weight of lambs before slaughter are of great importance in lamb production to obtain high quality lamb carcasses. However, the interest of this research was that feeding management plays an important role in feeding behavior, which leads to changes in plasma parameters and meat quality that are not well understood. This experiment will help provide more data and a better understanding of these relationships to support strategic feeding management to increase performance and improve carcass characteristics. Understanding how feeding management affects growth performance, feeding behavior, carcass characteristics, and meat quality may be important to implement strategic herd management aimed at increasing farm profitability. The objective of the study was to evaluate the effects of three production systems, nomadic grazing and different levels of concentrates, one with medium (roughage/concentrate ratio 50:50) and one with high concentrate (roughage/concentrate ratio 30:70) on growth performance, behavior, blood parameters, carcass characteristics, and meat quality of *longissimus dorsi* muscle (LD). We hypothesized that concentrate feeding would alter feeding behavior, improve growth performance, carcass characteristics, and directly affect lamb LD muscle quality.

## Material and methods

### Ethics statement

The study was carried in central Iran (Isfahan province, Iran) during the grazing season lasted 90 days (February 1, 2019 to May 1, 2019) using Turki-Ghashghaei breed sheep. All methods were performed according to Iranian Council of Animal Care regulations and the study complies with ARRIVE guidelines for reporting in vivo experiments (IACUC#2019/09.2).

### Animals and experimental design

The lambs had grazed exclusively with their mother before the experiment. The experiment started in February 2019, with an initial body weight (BW) 29.5 ± 4.15 kg (mean ± SD). A total of 30 male lambs (n = 10 lambs/treatment) were randomly assigned to three production systems, rotational grazing (NG, approximately 150 ha) and two different levels of concentrates (CON), one with medium (MC, roughage/concentrate ratio 50:50) and one with high concentrate (HC, roughage/concentrate ratio 30:70) during the 90-day fattening period. The NG lambs were reared in Aghdash (31°36′09″ N 51°32′32″ E) in Vardasht Rural District of Semirom City (Isfahan Province, Iran). The regions are botanically diverse upland grasslands with

dominant native species such as *Sorghum halepense*, *Prangos ferulaceae*, *Daphne mucronata Royle*, *Astragalus*, *Borage officinalis*, and wild barley grass communities. At the beginning of the experiment, all lambs averaged 90 ± 4 days of age (mean ± SD) and were slaughtered at an average of 180 ± 3 days of age (mean ± SD). In the NG group, animals were housed in loose pens and kept 14 hours per day (0500–1900 h) on green nomadic pastures under a rotational grazing system (rotational grazing is a system where a large pasture is divided into smaller paddocks allowing livestock to be moved from one paddock to the other easily) and stalled at night until the end of experiment. After a 14-d adaptation period, the remaining 20 animals were divided into two experimental groups that received different feeding until the 90-days final fattening period (Fig 1). Lambs with initial BW 30.2 ± 4.25 kg (mean ± SD) were fed MC indoors [roughage/concentrate ratio 50:50% (MC, n = 10)] and initial BW 30.2 ± 4.47 kg (mean ± SD) were fed HC indoors [roughage/concentrate ratio 30:70% (HC, n = 10)]. Salt blocks was always available to all animals. The lambs that were reared indoors in a single pen (1.2 m × 1.5 m). Each pen was equipped with a concrete feed bunk and clean water. As part of the harvesting process (Golchin Trasher Hay Co., Isfahan, Iran), alfalfa was chopped to a length of 30 mm by using a harvester with a screen size regulator. In the harvesting process, wheat straw was chopped to a theoretical cut length of 10 mm by using a harvester (Golchin Trasher Hay Co., Isfahan, Iran). Clean wood shavings and sand were used as bedding and replaced daily. All feeds that contained supplemental feeds were formulated according to NRC requirements for small ruminants [9] and all the animals able to access the individual trough (30 cm) at the same time. Thus, *ad libitum*-fed lambs (as total mixed rations) were allowed to leave 20% refusals of feed offered twice daily (0900 and 1700 h). During each feeding, orts were collected daily and weighed before feed was given to the animals, and the feed consumption was then measured. Representative samples were taken from each trough for further analysis.

All lambs were weighed at the beginning of the experiment and then weighed weekly. FCR was calculated as the ratio of average daily feed intake to average daily gain (ADG) 90 days. An ultrasound scanner (Ultrascan 50, Linear 3.5 MHz Scan, Alliance Medical Inc) was used to measure the thickness of backfat and loin fat in live lambs between the third and fourth lumbar vertebrae (halfway between the last rib and the hip bone). Fig 1 shows a graphical representation of the experimental design.

### Feed chemical analysis

The ingredients and their chemical composition for the different feed treatments are shown in Table 1. Samples of ingredients, subsamples of feeds, and individual refusals were collected immediately before the morning feeding during the 7-d collection period and placed into nylon bag stock. All samples were frozen immediately at −20 ˚C until analyses. After thawing, weekly feed and ort samples were treated at 60˚C in a forced-air oven (Memmert, Schwabach, Germany) for 48 h and then ground to a size less than 1 mm using a Wiley mill (Arthur H. Thomas; Philadelphia, PA, USA) for each individual lamb. Samples were analyzed in triplicate according to Association of Official Agricultural Chemists (AOAC) [10] for crude protein (CP) by the Kjeldahl method (Kjeltec 1030 Auto Analyzer, Tecator, Höganäs, Sweden; method 955.04), ash (method 942.05), ether extract (method 920.39), and neutral detergent fiber (NDF) using an ANKOM$^{200}$ Fiber Analyzer (A200 model, Ankom Technology, Macedon, NY) according to the manufacturer's recommended reagents and filter bags (#F57) with pore size of 25 μm. Ash content was measured after combustion in a muffle furnace at 550˚C overnight. The content of non-structural carbohydrates in the feed was calculated on a dry matter basis [9] as follows: 100− (NDF%+CP%+ether extract%+ash%).

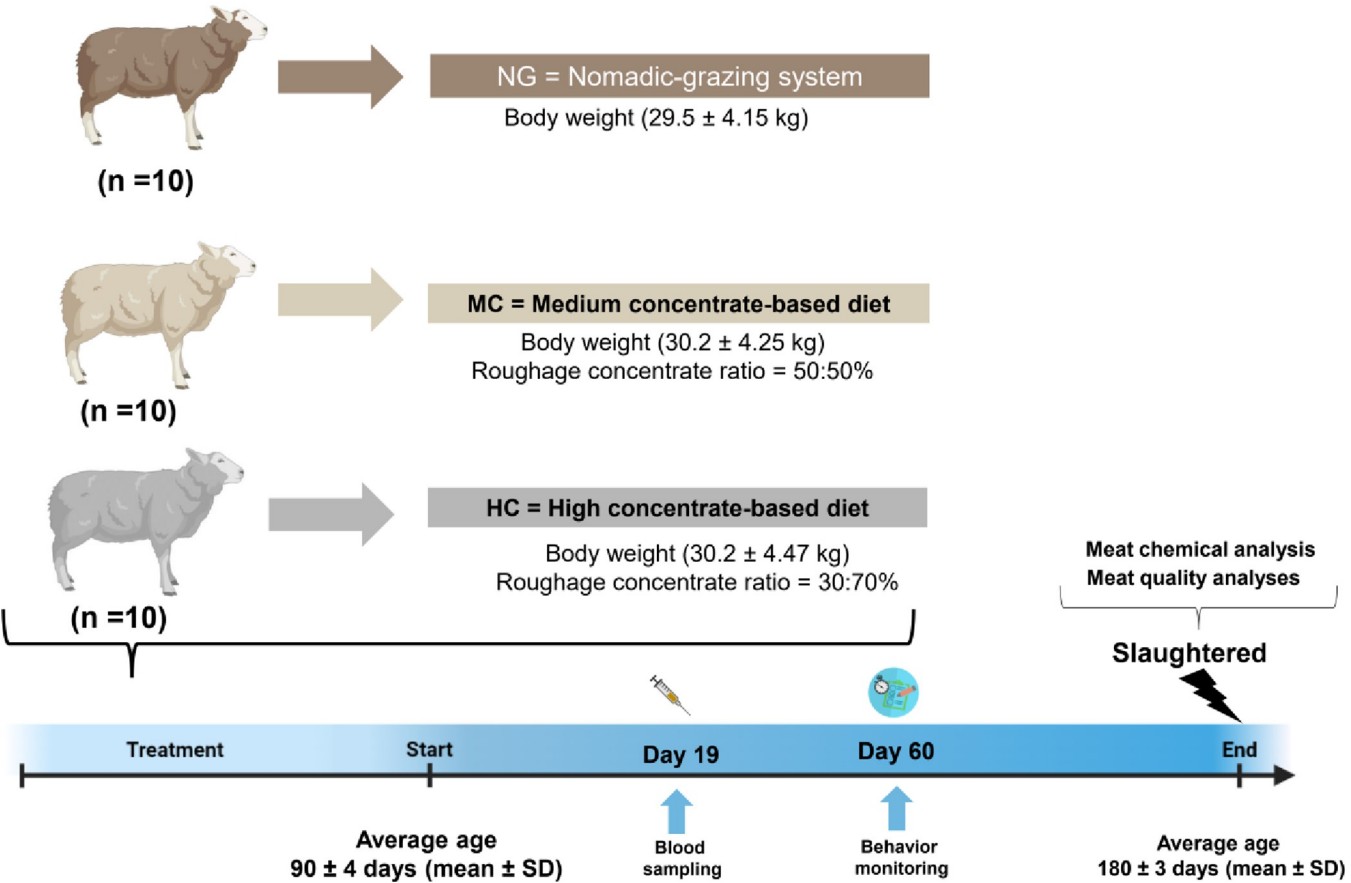

**Fig 1. Overview of the animal experiment.** The objective of the study was to evaluate the effects of three production systems on growth performance, behaviour, chemical characteristics, carcass, and meat quality. A total of 30 lambs (n = 10 lambs/treatment) were randomly assigned to three production systems that included nomadic grazing (NG) and two different levels of concentrates (CON) one with medium (MC, 50% concentrate) and another with high concentrate (HC, 70% concentrate). At the beginning of the experiment, all lambs averaged 90 ± 4 days of age (mean ± SD) and were slaughtered at an average of 180 ± 3 days of age (mean ± SD). The Figure was designed by BioRender online software (https://app.biorender.com).

## Blood collection

Nineteen days after the start of the experiment, blood samples were collected by venipuncture in the coccygeal vein 4 h after the start of grazing in the NG group and 4 hours after feeding in the groups fed concentrates. Samples were collected in 10 mL serum clot activator evacuated tubes. Tubes were centrifuged within 30 minutes (min) of collection at $2500 \times g$ for 20 min to separate the serum layer, which was then stored in aliquots at −20°C until further analysis. Glucose, cholesterol, triglyceride (TG), high-density lipoprotein cholesterol (HDL), urea, aspartate aminotransferase (AST), alanine aminotransferase (ALT), lactate concentrations, and total antioxidant capacity (TAC) were measured in the serum samples using an autoanalyzer (Abbott Alcyon 300; Abbott Laboratories; Chicago, IL, USA) with commercial kits (Pars Azmoon Co.; Tehran, Iran) according to the manufacturer's instructions. Serum β-hydroxybutyrate (BHB) was determined by Randox colorimetric kits (Randox Laboratories Ltd.; Ardmore, UK). Phosphorus (P) and calcium (Ca) concentrations in the serum were determined using flame atomic absorption spectrophotometry (UNICCO, 2100; Zistchemi Co.; Tehran, Iran).

**Table 1. Ingredients and chemical composition of experimental diets.**

| Item | Treatments | |
|---|---|---|
| | MC | HC |
| *Ingredient*, % DM | | |
| Alfalfa hay | 33.40 | 20 |
| Wheat straw | 16.60 | 10 |
| Barley | 28.2 | 39.48 |
| Corn | 10.1 | 16.45 |
| Wheat bran | 3.85 | 5.39 |
| Soybean meal | 6.75 | 7.21 |
| Calcium carbonate | 0.09 | 0.441 |
| Oxid manganese | 0.09 | 0.127 |
| White salt | 0.220 | 0.224 |
| Urea | 0.225 | 0.224 |
| Mineral premix[1] | 0.224 | 0.224 |
| Vitamin premix[2] | 0.266 | 0.266 |
| *Chemical composition* | | |
| Dry matter, % of fresh feed weight | 95.67 | 94.67 |
| Organic matter, % DM | 98.35 | 98.58 |
| Crude protein, % DM | 15.75 | 16.53 |
| Ether extract, % DM | 2.44 | 3.34 |
| Neutral detergent fiber, % DM | 42.16 | 34.79 |
| Ash, % DM | 8.12 | 8.03 |
| Non-fiber carbohydrate, % DM | 27.2 | 31.98 |
| *Metabolizable energy, Mcal/kg DM* | 2.33 | 2.57 |

[1]Mineral mix contains 4,040 ppm Cu, 20,000 ppm Mg, 12,200 ppm Mn, 282,000 ppm Ca, 16,200 ppm Zn, 105 ppm Co, 190 ppm I, and 80 ppm Se.

[2]Vitamin mix contains 800,000 ppm vitamin A, 150,000 ppm vitamin D3, 5,000 ppm vitamin E.

[3]NFC was calculated as DM − (NDF% + CP%+ ether extract% + ash%) [9].

## Lamb behavior

Eating (chewing feed in the mouth), rumination (chewing regurgitated food while standing or lying down), drinking (swallowing water), standing (without moving or exhibiting any other behavior such as eating or ruminating), and resting (lying/sleeping without exhibiting any other behavior such as eating or ruminating) activities of the lambs in all treatments were visually assessed by eight trained observers (one every 3 h) on d 60 over 24 h (0730 h). Lamb activity was recorded every 5 min and behavior was assumed to continue throughout the 5 min. All lambs were marked with different colors on their bodies to record their activity.

## Slaughtering and sampling

At the end of the experimental period, the designated veterinarian certified that the animals did not suffer any adverse effects of the procedures. When reach the target day, lamb were individually weighed and transported to the slaughterhouse located in the Animal Science department of the Isfahan University of Technology. The personnel of the slaughterhouse were informed of the procedure in advance. Empty body weight (EBW) and hot carcass weight (HCW) were determined for each lamb at the time of slaughter. Slaughter was carried out following the halal method [11]. Weights of the liver, gastrointestinal tract (GIT) including rumen, small and large intestine, and fat tail were determined as a percentage of BW pre-

slaughter. Gastrointestinal tract was flushed with water and drip dried before being weighed. *Longissimus dorsi* muscle was excised from the right half carcass and sampled for the following meat quality determinations immediately after slaughter.

## Meat quality analyses

The pH of LD muscle was measured at a depth of approximately 2.5 cm in the muscle on the 5/6 rib of the right half of the carcass at hourly intervals between 0 and 24 h *post mortem* using a pH probe (Jenoy 3330, Vernon Hills, USA) equipped with a penetrating glass electrode and a thermometer. The color of a meat cut (2.5 cm-thick) from the thirteenth thoracic vertebra, was determined at 0 h *post mortem* (immediately after slaughter) and 24 h *post mortem* in the same slice. Meat color parameters, including lightness ($L^*$), redness ($a^*$), and yellowness index ($b^*$) were determined using the Nippon Denshoku colorimeter (Lurton, RGB 1002, Taiwan) with D 65 light source, 10-mm-diameter sample stage, and standard observation angle of 10˚. Texture profile analyses were performed using a Texture Analyzer (SANTAM STM 1, Tehran, Iran) equipped with a 1 kg load cell and a needle-shaped probe (0.5 cm diameter). The device was set and moved perpendicular to the 2 cm thick and 4×4 cm$^2$ sample at 10 mm/s speed and 0.05 N trigger force point. The samples were compressed twice to 25% of their original thickness [12]. The texture variables, including hardness (maximum force required to compress the sample), gumminess (the force required to disintegrate a semi-solid sample for swallowing), cohesiveness (the extent to which the sample can be deformed before rupture), chewiness (the work required to masticate the sample for swallowing), and springiness (the ability of the sample to recover its original shape after removing the deformation force) were calculated as described previously [13]. For drip loss determination, a sample of 80 and 100 g from a nylon bag was stored in a refrigerated room (4˚C) for 24 h, and other samples were stored in a freezer (-16˚C) for 1, 2, 3, and 4 months. At each target time, samples were removed from the containers, blotted dry, and weighed using the procedure described previously [14]. loin At each target time, samples were removed from the containers, blotted dry, and weighed using the procedure described previously [14]. Once weighted samples were returned to the container and, at each target time, percent drip loss was calculated as follows:

$$Drip\ loss\ \% = \frac{(W_{br} - W_{ar})}{W_{br}} \times 100$$

Where $W_{br}$ and $W_{ar}$ represent the weight of the meat samples before refrigeration (initial weight) and after surface water elimination at each target time, respectively.

To measure cooking losses (%), weighted LD muscle meat from the ninth to tenth rib of the right half of the carcass was placed in plastic bags and cooked in a water bath at 75˚C for 25 min until the center of the sample reached 72˚C, as described previously [15]. The samples were removed from their respective bags, dried with paper towels, and reweighed. Cooking loss (%) was calculated as the percent weight loss of the cooked sample from the initial sample weight. Lipid oxidation in the LD muscle samples was measured by determining thiobarbituric acid-reacting substance (TBARS) levels 24 h after slaughter through measuring mg malonaldehyde/kg sample after 0-, 1-, 2-, and 3-month frozen storage. Briefly, 20 g meat was mixed with 50 mL 20% trichloroacetic acid (TCA) for 2 min. The blender contents were rinsed with 50 mL water, mixed, and filtered through a Whatman #1 filter. Then, 5 mL TCA extract was mixed with 5 mL 0.01 M 2-thiobarbituric acid, incubated at 100˚C for 1 h, and again filtered through a Whatman # 1 filter. The absorbance of the pink solution was measured at 532 nm using a UV/vis spectrophotometer. TBARS was expressed as mg malonaldehyde/kg sample [16].

## Meat quality and texture characteristics

Moisture, CP, fat, and crude ash content were analyzed using previously published protocols [17]. The meat was cut into thin pieces and immediately frozen at −18˚C. To measure the fat content, the meat was dehydrated using a freeze dryer (FD-5003- BT; Dena vacuum; Tehran; Iran) and the fat was extracted from the dried meat using the Soxhlet method [15]. All analyzes were performed in triplicate. For protein solubility analysis, LD muscle between the sixth and thirteenth ribs of the right half of the carcass was separated and thawed at 4˚C. A portion of the minced meat was mixed with four volumes of cold extraction medium comprising 100 mM KCl, 25 mM $K_2HPO_4/KH_2PO_4$, and 2 mM EDTA (pH = 7.5) and homogenized at 16,000 $g$ for 30 s (Ultra-Turraxt 18 IKA, Burladingen, Germany). Subsequently, the homogenate was centrifuged at 16,000× $g$ for 15 min. After separating the supernatant containing the sarcoplasmic proteins, the resulting sediments were homogenized again with four volumes of the extraction buffer and then passed through a polyethylene strainer (18 mesh) for separating the stromal proteins. The resulting solution was centrifuged at 16,000× $g$ for 15 min and the sediments were homogenized with four volumes of extraction buffer containing 10% NaCl. The protein concentration of the solution was determined by the Biuret method [18] and adjusted to 3% by adding the NaCl extraction buffer. To determine the solubility of the myofibrillar protein, 1 mL 3% protein solution was mixed with 2 mL 10% saline–phosphate buffer (pH = 7.3). The solution was stirred for 30 min and then centrifuged at 10,000 × $g$ for 15 min. The supernatant was then extracted and its protein concentration was determined by the Biuret method. The absorbance of the protein solution was measured at 540 nm (Unico UV2100, Palm, UK) and expressed as mg protein/mL solution. Bovine serum albumin (Sigma Chemical Co.; St. Louis, MO, USA) was used to construct the standard curve. Myofibrillar protein solubility was determined by determining the difference between total and sarcoplasmic protein solubility. TAC was measured spectrophotometrically at 520 nm and expressed as mg tissue protein increasing the optical density by 0.01 per minute at 37˚C. Data for TAC were expressed as specific activity units per mg protein (U/mg protein) in meat. Assays were performed using assay kits obtained from Nanjing Jiancheng Institute of Bioengineering (Nanjing, People's Republic of China) according to the manufacturer's instructions.

## Statistical analysis

All data were screened for normality using PROC UNIVARIATE and normalized as required using a $log_{10}$ transformation, but estimates are back-transformed to the response scale. Data with repeated measures (color of LD muscle) were analyzed with PROC MIXED from SAS (SAS 9.1; SAS Institute Inc; Cary, NC, USA) when variables were measured over time. The model consisted of treatment (NG, MC, and HC), time, and treatment × time as fixed effects and lamb as a random effect. The lowest level of Bayesian information criterion (fit statistic) was used to select the covariance structure and the compound symmetry structure was modeled accordingly. The initial BW was used as a covariate in the BW model. Data on body condition, behavioral parameters, blood parameters, and carcass and LD muscle characteristics were analyzed using the same model without the time effect. A PCA was also performed using the Factor Procedure. The variables included were productive parameters (initial BW, pre-slaughter BW, and ADG), body condition (loin and backfat thickness), behavioral parameters (eating, ruminating, drinking, resting, and standing), blood parameters (glucose, cholesterol, TG, HDL, urea, BHB, AST, ALT, P, Ca, lactate level, and TAC), non-carcass parts (liver and GIT weight), carcass characteristics (EBW, HCW, and fat tail weight), and viscera meat quality characteristics (moisture, crude protein, fat, and crude ash percentage, initial and final pH, initial and 24 h *post mortem* $L^*$, $a^*$, $b^*$; cooking losses, and TAC), texture parameters (hardness,

springiness, cohesiveness, gumminess, and chewiness), myofibril and sarcoplasm protein content, TBARS and drp loss, and Partial least squares-discriminant analysis (PLS-DA) identified differential variables between experimental groups, while ranking variables by importance in the projection (VIP) scores according to the importance of variables in discriminant groups. The permutation test used maximum 2000 permutations with the separation distance test to evaluate the significance of the class discrimination determined by PLS-DA. Classification and cross-validation were performed with maximum five components. All raw data from the measured parameters are available in S1 Table.

## Results

### Intake, growth performance, and body condition

Feed intake and FCR of CON fed lambs are shown in Table 2. These values were not affected by the levels of concentrate. Animal growth performance and body condition are also shown in Table 2. Initial BW of lambs did not differ between treatments, but pre-slaughter BW tended to be lower for NG lambs compared CON-fed lambs ($P<0.10$). In addition, CON-fed lambs had higher ADG than NG lambs ($P<0.05$). Further, the feeding system significantly affected loin thickness ($P<0.05$), but did not affect backfat thickness measured before slaughter. Loin thickness was significantly higher in CON than NG, but, no significant differences were found between MC and HC.

### Feeding behavior

The NG lambs spent more time eating, drinking, and standing, but less time ruminating and resting than CON-fed lambs ($P<0.05$, Fig 2A–2E).

### Plasma parameters

Plasma cholesterol, TG, and HDL concentrations were higher in NG lambs than in CON fed lambs ($P<0.05$; Table 3). Plasma concentrations of urea and BHB were higher in NG-fed lambs than in CON-fed lambs ($P<0.05$). The MC-fed lambs had higher plasma urea and BHB concentrations than HC-fed lambs ($P<0.05$). Plasma P concentrations were lower in NG -fed lambs than in CON-fed lambs ($P<0.05$). Plasma concentrations of glucose, ALT, and TAC did not differ between production systems, but AST and lactate concentrations tended to be higher in HC-fed lambs than in MC-fed lambs ($P<0.10$).

**Table 2. Intake, growth performance, and body condition of lambs from different production systems.**

| Item | Treatments | | | SEM | Contrast *P*-Value | | |
|---|---|---|---|---|---|---|---|
| | NG | MC | HC | | NG vs. CON | MC vs. HC | |
| Feed intake, kg DM/day | – | 1.60 | 1.63 | 0.076 | – | 0.859 | |
| FCR, kg of DM/kg ADG | – | 7.77 | 7.00 | 0.312 | – | 0.271 | |
| *Growth performance* | | | | | | | |
| Initial BW, kg | 28.32 | 31.15 | 30.87 | 0.843 | 0.157 | 0.875 | |
| Pre-slaughter BW, kg | 41.80 | 46.48 | 47.70 | 1.152 | 0.070 | 0.308 | |
| ADG | 0.19 | 0.21 | 0.23 | 0.008 | 0.035 | 0.308 | |
| *Body condition, mm* | | | | | | | |
| Loin thickness | 21.15 | 23.80 | 22.61 | 0.492 | 0.039 | 0.320 | |
| Fat thickness | 2.25 | 2.42 | 2.31 | 0.065 | 0.393 | 0.512 | |

ADG = average daily gain, BW = body weight, NG = nomtic-grazing system, MC = medium concentrate-based diet, HC = high concentrate-based diet, CON = concentrate-based diet including both, MC and HC), FCR = feed conversion ratio, SEM = standard error of the mean.

## Carcass characteristics

Non-carcass parts and characteristics are shown in Table 4. At the end of the rearing period, CON-fed lambs had lower GIT and fat tail ($P<0.01$) and higher EBW ($P<0.05$) and HCW ($P<0.01$) than NG lambs.

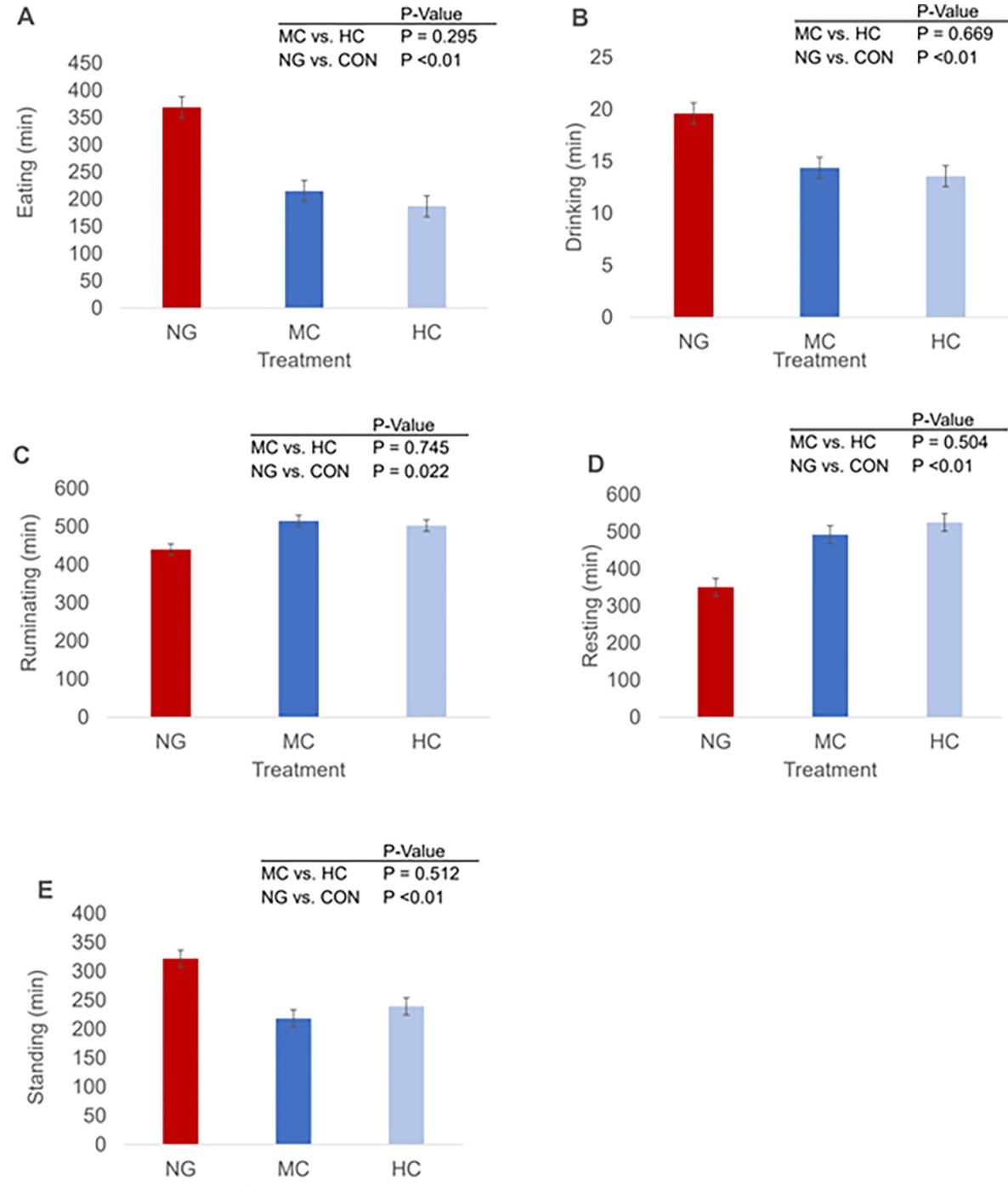

**Fig 2.** Lambs eating (A), drinking (B), ruminating (C), resting (D), and standing (E) behaviour by production system: NG = nomadic-grazing system, HC = high concentrate-based diet, MC = medium concentrate-based diet, CON = concentrate-based diet (including both HC and MC). Bars in each column represent the standard error of the mean.

**Table 3. Plasma parameters of lambs from different production systems.**

| Item | Treatments | | | SEM | Contrast P-Value | |
|---|---|---|---|---|---|---|
| | NG | MC | HC | | NG vs. CON | MC vs. HC |
| *Plasma parameter* | | | | | | |
| Glucose, mg/dL | 87.50 | 89.25 | 89.37 | 1.056 | 0.425 | 0.964 |
| Cholesterol, mg/dL | 53.15 | 43.69 | 40.37 | 1.959 | <0.01 | 0.378 |
| TG, mg/dL | 39.70 | 20.94 | 25.12 | 2.750 | <0.01 | 0.428 |
| HDL, mg/dL | 36.45 | 22.67 | 24.29 | 1.814 | <0.01 | 0.493 |
| Urea, mg/dL | 46.85 | 41.56 | 34.50 | 1.470 | <0.01 | 0.017 |
| BHB, mmol/L | 0.50 | 0.49 | 0.30 | 0.030 | 0.054 | 0.007 |
| AST, IU/L | 116.6 | 116.6 | 129.5 | 2.739 | 0.228 | 0.067 |
| ALT, IU/L | 23.00 | 22.93 | 22.12 | 0.758 | 0.771 | 0.697 |
| Phosphorus, mg/dL | 7.54 | 8.02 | 8.30 | 0.139 | 0.028 | 0.413 |
| Calcium mg/dL | 11.48 | 12.02 | 11.73 | 0.118 | 0.102 | 0.341 |
| Lactate, mmol/L | 2.46 | 2.18 | 2.92 | 0.166 | 0.781 | 0.095 |
| TAC, mmol/L | 0.43 | 0.42 | 0.39 | 0.013 | 0.309 | 0.332 |

NG = nomadic-grazing system, HC = high concentrate-based diet, MC = medium concentrate-based diet, CON = concentrate-based diet (including both HC and MC), ALT = alanine aminotransferase, AST = aspartate aminotransferase, BHB = β-hydroxybutyrate, HDL = high-density lipoprotein cholesterol, SEM = standard error of the mean, TAC = total antioxidant capacity, TG = triglyceride.

## Meat quality and texture characteristics

The chemical composition and meat quality parameters are shown in Table 5. Chemical composition of the meat did not differ between the three groups. The diets affected the pH of the meat immediately after slaughter ($P<0.05$), but did not affect the pH 24 h *post mortem*. Meat from the NG lambs had higher $L^*$ ($P<0.05$), lower $a^*$ ($P<0.01$), and higher $b^*$ ($P<0.01$) index than that from the CON-fed lambs at the start of blooming. However, meat from NG-fed lambs had higher $L^*$ ($P<0.01$) and tended to have higher $a^*$ ($P<0.10$) than that from CON-fed lambs 24 h *post mortem*. In addition, $a^*$ tended ($P<0.01$) to increase slightly in the HC-fed lamb meat than that in MC-fed lamb meat 24 h *post mortem*. Textural variables (hardness, springiness, cohesiveness, gumminess, and chewiness), cooking losses, TAC in muscles, and muscle protein myofibrillar and sarcoplasm did not differ among the three groups. However, drip loss and TBARS level did not differ between the meat of NG and CON-fed animals; instead, as expected, drip loss and lipid oxidation ($P<0.01$) increased with the duration of cold storage (Fig 3A and 3B).

**Table 4. Carcass characteristics (kg or % of BW pre-slaughter) of lambs from different production systems.**

| Item | Treatments | | | SEM | Contrast P Value | |
|---|---|---|---|---|---|---|
| | NG | MC | HC | | NG vs. CON | MC vs. HC |
| *Non-carcass parts* | | | | | | |
| Liver weight, % of BW pre-slaughter | 1.61 | 1.40 | 1.53 | 0.040 | 0.077 | 0.197 |
| GIT weight, % of BW pre-slaughter | 6.18 | 5.52 | 5.43 | 0.107 | <0.01 | 0.695 |
| *Carcass characteristics* | | | | | | |
| HCW, kg | 21.42 | 28.11 | 27.99 | 0.955 | 0.002 | 0.886 |
| HCW, % of BW pre-slaughter | 49.94 | 56.89 | 55.29 | 0.935 | <0.01 | 0.395 |
| Fat tail weight, % of BW pre-slaughter | 6.09 | 9.84 | 9.50 | 0.508 | <0.01 | 0.730 |

BW = body weight, NG = nomadic-grazing system, MC = medium concentrate-based diet, HC = high concentrate-based diet, CON = concentrate-based diet including both, MC and HC), EBW = empty body weight, GIT = gastrointestinal tract (stomach, small intestine, and large intestine), HCW = hot carcass weight, SEM = standard error of the mean.

**Table 5. *Longissimus dorsi* muscle characteristics of lambs from different production systems.**

| Item | Treatments | | | SEM | Contrast *P*-Value | |
|---|---|---|---|---|---|---|
| | NG | MC | HC | | NG vs. CON | MC vs. HC |
| *Chemical composition, %* | | | | | | |
| Moisture | 63.87 | 64.43 | 65.56 | 0.520 | 0.308 | 0.416 |
| Crude protein | 20.85 | 21.46 | 21.12 | 0.361 | 0.581 | 0.729 |
| fat | 5.73 | 5.90 | 6.57 | 0.385 | 0.542 | 0.528 |
| Crude ash | 1.09 | 1.11 | 1.16 | 0.015 | 0.137 | 0.243 |
| *pH (h post mortem)* | | | | | | |
| pH $_{0\,h}$ | 7.23 | 7.46 | 7.48 | 0.041 | 0.004 | 0.826 |
| pH $_{24\,h}$ | 5.78 | 5.79 | 5.78 | 0.017 | 0.818 | 0.911 |
| *Color parameters at 0 h* | | | | | | |
| Lightness index ($L^*$) | 46.11 | 45.71 | 45.77 | 0.076 | 0.020 | 0.741 |
| Redness index ($a^*$) | 20.03 | 21.01 | 20.94 | 0.158 | <0.01 | 0.843 |
| Yellowness index ($b^*$) | 9.84 | 9.52 | 9.52 | 0.049 | <0.01 | 0.961 |
| *Color parameters at 24 h* | | | | | | |
| $L^*$ | 53.67 | 52.86 | 53.09 | 0.098 | <0.01 | 0.216 |
| $a^*$ | 19.21 | 18.49 | 19.03 | 0.126 | 0.063 | 0.087 |
| $b^*$ | 11.96 | 12.06 | 12.26 | 0.103 | 0.360 | 0.469 |
| *Texture profile analysis* | | | | | | |
| Hardness, N | 75.02 | 54.04 | 50.62 | 8.068 | 0.499 | 0.681 |
| Springiness, m | 0.57 | 0.56 | 0.52 | 0.011 | 0.142 | 0.100 |
| Cohesiveness | 0.46 | 0.46 | 0.42 | 0.013 | 0.400 | 0.206 |
| Gumminess, N | 29.61 | 31.74 | 26.22 | 3.068 | 0.928 | 0.483 |
| Chewiness, J | 16.59 | 14.86 | 14.27 | 1.675 | 0.874 | 0.842 |
| *Cooking losses, %* | 32.67 | 33.55 | 34.67 | 1.150 | 0.561 | 0.715 |
| *Myofibrils, mg/mL* | 16.73 | 18.39 | 17.93 | 0.539 | 0.120 | 0.753 |
| *Sarcoplasm, mg/mL* | 25.34 | 28.99 | 25.61 | 1.069 | 0.371 | 0.245 |
| *TAC, mmol* | 0.73 | 0.82 | 0.79 | 0.025 | 0.126 | 0.637 |

NG = nomadic-grazing system, HC = high concentrate-based diet, MC = medium concentrate-based diet, CON = concentrate-based diet (including both HC and MC), SEM = standard error of the mean, TAC = total antioxidant capacity.

## Multivariate analysis

Fig 4 shows the relationships between blood parameters, feeding behavior, and meat quality of lambs. Principal component analysis (PCA, Fig 4A) and partial least squares discriminant analysis (PLS-DA, Fig 4B) based on blood parameters, feeding behavior, and meat quality showed complete and significant separation of NG and CON-fed lambs, but overlap between MC- and HC-fed lambs. Validation plots showed that the PLS-DA model was valid based on separation distance (Fig 4C, *P*< 5e-04) using a different number of components (Fig 4D). According to the VIP score using the above PLS-DA model (Fig 4E), feeding time, yellowness, and redness index after slaughter were the variables that contributed most to the separation of treatment groups.

## Discussion

### Feed intake, FCR, and animal performance

In this study, no differences were found between the feeding strategies of MC and HC lambs. Moreover, FCR of HC- and MC-fed lambs was also similar. Similarly, no differences were found in FCR by Jabbar and Anjum [19] after feeding Lohi lambs with concentrate (forage to

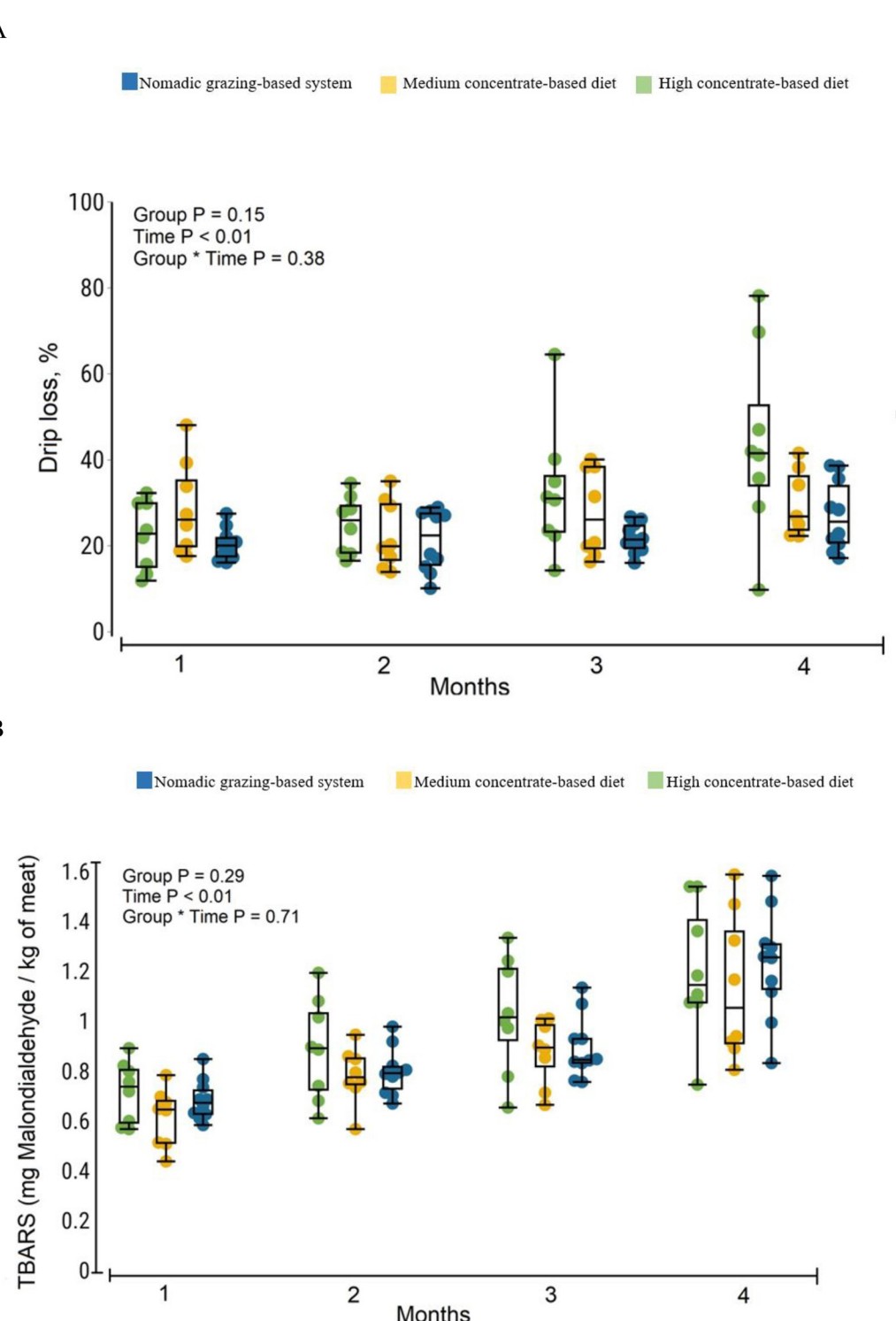

**Fig 3.** (A) Drip loss (%) and (B) Thiobarbituric acid reacting substances (TBARS) formation (mg malondialdehyde per kg meat) in M. *longissimus dorsi* in lambs fed nomadic-grazing based system, medium concentrate-based diet, and high concentrate-based diet (n = 10 lambs/treatment).

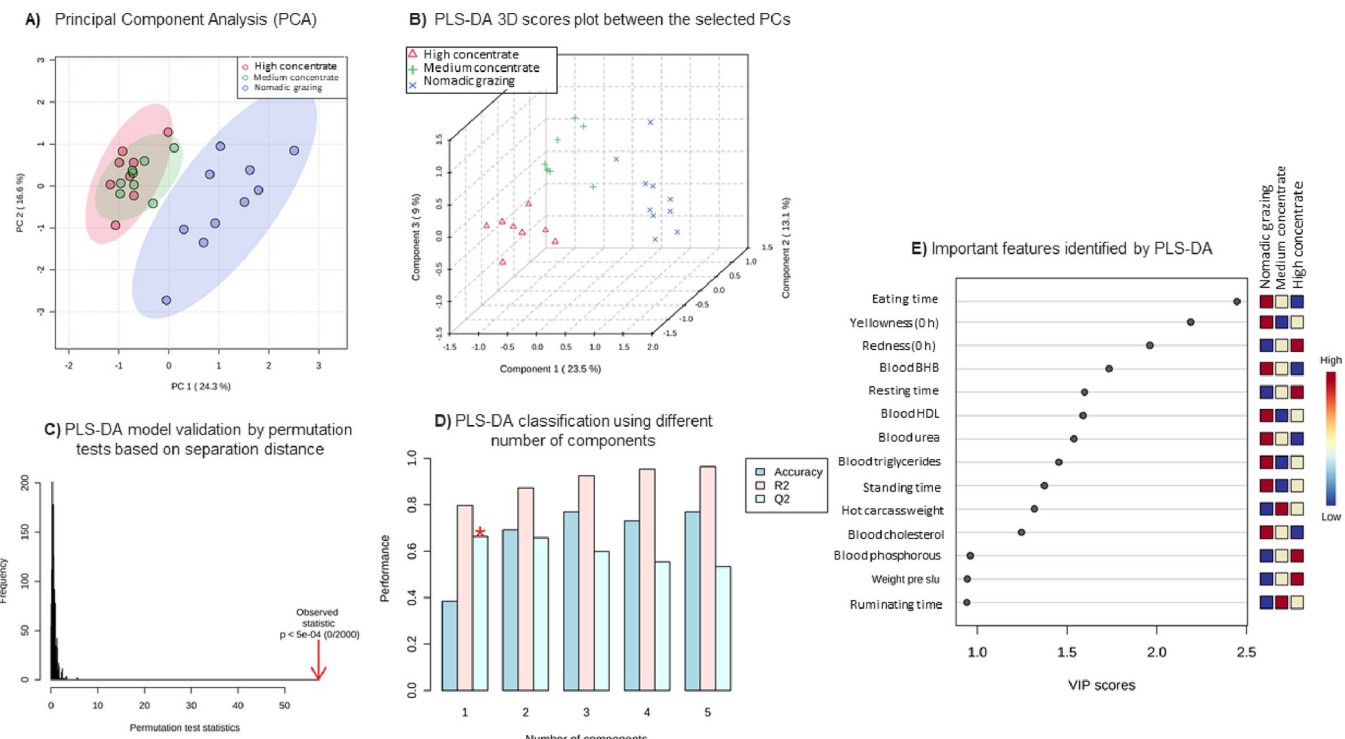

**Fig 4.** A. Principal component analysis (PCA), B. 3D scores plot of partial least squares-discriminant analysis (PLS-DA), C. PLS-DA model validation by permutation tests based on separation distance (P < 5e-04 (0/2000), D. PLS-DA classification using a different number of components (the red star indicates the best classifier), E. Important features identified by PLS-DA of lambs in the nomadic grazing-based system (NG, n = 10), medium concentrate-based diet (MC, n = 10), and high concentrate-based diet (HC, n = 10). The colored boxes on the right indicate the relative concentrations of the corresponding variable in each group under study.

concentrate ratios 50:50 and 25:75). In this study, lambs fed CON had higher ADG at the end of the growth period than those fed NG. Gallo et al. [20] reported that the lower feed intake associated with higher weight gain and carcass yield was due to the higher available energy in the feed. The loin thickness of NG lambs was significantly lower than that of CON -fed lambs. Our result is in agreement with that of Borton et al. [21], whose observations are probably due to the partitioning of energy for tissue growth, since tissue maturation occurs in the order of bone, lean meat and fat.

## Feeding behavior

The welfare of farm animals can be improved by increasing their behavioral repertoire, including the maintenance of behaviors such as resting, standing, walking, feeding, and drinking [22]. The effect of feeding can be confounded with different levels of physical activity. The effect of feeding can be confounded with different levels of physical activity (Fig 2A–2E). The PCA plot shows that the NG lambs are different from the CON fed lambs. According to the VIP scores of the above PLS-DA model, eating time was selected as the highest ranked variable. The proportion of time spent eating and standing was greater in NG lambs than in CON-fed lambs, indicating physical activity. Based on feed availability, Salem et al. [23] found that the time spent grazing herbaceous vegetation actually corresponds to the time of searching vegetation residues above ground. Rumination time is a good marker of feeding behavior to discriminate between rumen function and animal welfare [24]. In this study, lambs fed CON spent more time rumination. Interestingly, behavioral changes associated with feeding behavior in

ruminants may influence the ability to cope with changes in the nutritional environment [25]. In this study, drinking and standing times were higher in NG lambs, but resting times were lower than in CON fed lambs, which was related to foraging and more standing times. It is likely that the indoor groups ate more than the outdoor group. In addition, the longer drinking time could be an indication of a drier diet. Charlton et al. [26] reported that > 70% of water was consumed during the day and drinking correlated with feeding and milking. Animals benefit from resting time, which indicates their welfare. Therefore, when comparing extensive/outdoor and intensive/indoor beef production systems, it is generally assumed that beef in an extensive system have more space on average [27] and opportunistically engage in higher levels of physical activity [8].

## Blood parameters

Measurement of blood metabolites allows determination of discrete differences in energy status [28]. Most diseases are influenced by certain changes in the concentration of blood parameters. NG lambs had higher cholesterol, triglyceride, and HDL levels than the CON-fed lambs, which could be due to the difference in diet. In addition, HDL helps prevent the narrowing of arterial walls by removing excess cholesterol and transporting it to the liver for excretion [29]. The HDL concentration increased in NG lambs, which was confirmed by the change in blood cholesterol concentration between treatments observed in our study. In addition, this study showed that CON had no effect on liver enzymes (AST and ALT) and thus did not appear to have negative effects on the function of organs related to blood substances. In this study, NG lambs had higher urea and BHB concentrations than CON-fed lambs. The increase in blood urea in NG lambs suggests that glucose levels may have been maintained by the increased use of non-essential AA for gluconeogenesis. In the absence of glucose, the concentrations of ketone bodies and urea produced by fat and protein catabolism increase. Moreover, urea and BHB concentrations were higher in MC-fed lambs than in HC-fed lambs. The higher protein content of alfalfa would explain the higher plasma urea in MC lambs. Serment et al. [30] also reported that goats fed high-concentrate diets (70% concentrate on dry matter basis) before feeding had lower plasma ammonia-N and urea concentrations than goats fed low-concentrate diets (35% concentrate on dry matter basis). Blood P concentrations were affected by the treated diets and were higher in lambs fed CON than in those fed NG. Similarly, switching from coarse to fine ground diet [31] and less digestible (52%) hay to more digestible (84%) concentrate [32] resulted in increased urinary phosphorus excretion, decreased salivary phosphorus excretion, and decreased rate at which phosphorus was removed from plasma via saliva.

## Carcass characteristics

The EBW and HCW were influenced by finishing system (Table 4) and were higher in CON-fed lambs than in NG lambs. Claffey et al. [33] reported that the difference in slaughter weight was likely due to the high energy intake and resulting high growth rate, which from a commercial perspective highlights the importance of finishing diets low in indigestible fiber to achieve maximum live weight gain and ultimately high carcass weight. Our data show that lambs fed CON had lower GIT weight than those fed NG. The rumen passage rate of legumes is usually higher than that of grasses because legumes pass through the rumen easily due to particle breakdown [34]. This may be attributed to the fractional rumen passage rate, as grasses may decrease the fractional rumen passage rate and increase the amount of undigested feed reaching the abomasum, which could increase the weight of abomasum tissue [35]. It is important to note that under natural rearing conditions, especially when young ruminants are kept on

pasture, most of the nutrients and stimulants for rumen development are obtained from fresh forage [36]. Large particles, fresh forage, and high fiber sources provide physical stimuli that increase rumen motility, masculinization, and rumen volume in calves [36]. The NG lambs had lower fat tail weights than lambs in other production systems. In the study by Claffey et al. [33], the increased energy content of the diet with increasing feeding of concentrates in a 50:50 ratio (0.9 and 3.0 kg concentrates and silage, respectively) could also help explain the difference in fat cover. We found similar performance and carcass quality in lambs fed MC and HC. There is much debate about whether MC is an attractive option for finishing lambs or for maintaining lamb growth when concentrate prices are relatively high compared to lamb. On a commercial level, this could be beneficial in situations where concentrate prices are relatively high compared to lamb meat.

## Meat quality and texture characteristics

The absence of differences in percent moisture content of meat between lambs from different feeding treatments is consistent with previous reports [37]. Neither crude protein nor fat content of the meat was affected by the treatments (Table 5). This study showed that initial muscle pH was higher in lambs fed CON than in lambs fed NG. The final pH did not differ between the different fattening strategies and was within the normal range in three groups. Color and color stability can be influenced by feeding management in extensive (grass/pasture) or intensive (concentrate/grain) systems [38]. Muscle color is an important criterion for evaluating meat quality and acceptability [38]. According to the VIP results of the above PLS-DA model, post-slaughter yellowness and redness indexes were selected as the second and third highest variables. NG lamb meat had higher $L^*$, lower $a^*$, and higher $b^*$ value than CON-fed lamb meat, which is consistent with previous results reported by Luo et al. [39], showing higher $b^*$ and lower $a^*$ value in the outdoors group than the other group of sheep's kept in one pen on the farm. They indicated that endurance exercise affected the muscle color of Sunit sheep, making muscle more yellow and less red. In our study, the $L^*$ of the meat from all lambs was above 40 and 45 after 0 and 24 h, respectively, indicating light-colored and acceptable meat. According to Velasco et al. [40], meat with $L^* \geq 34$ was acceptable on average and $L^* > 44$ was accepted by 95% of consumers. The color stability of meat was affected during refrigeration. In this sense, $L^*$ value increased at 24 h *post mortem* in NG lamb meat. The color of meat was influenced by *post mortem* factors, which essentially include refrigerated storage of meat, often referred to as aging [41]. Denaturation of proteins during rigor mortis increases the loss of free water and juice, resulting in changes in light scattering [42]. Sabow et al. [43] reported that $L^*$ value had a positive correlation with storage days' *post mortem*. Texture profile analysis, and cooking loss revealed no significant differences due to the finishing systems. In our study, the same hardness and springiness could be due to the same fat content in the meat between all dietary groups. Del Campo et al. [44] reported that several external factors such as feed, growth rate before slaughter, age of the animals at slaughter, and length of finishing period could influence the tenderness of the meat. The results of these experiments suggest that feeding is probably not the primary cause of the differences in texture profile studied. In this study, all lambs were treated identically during transport to pre-slaughter to minimize the negative effects of post-slaughter stress and glycogenolysis on animal meat quality. Myofibrillar and sarcoplasmic protein levels were similar in CON-fed lamb and those in NG lamb meat. Feeding pattern is known to affect the antioxidant status of meat [45], but our data showed that feeding CON or NG had no significant effect on TAC content. Accordingly, our result suggests that fattening strategy did not affect drip loss, but the amount of drip loss generally increased with increasing cold storage time (Fig 3A), which is consistent with the reports of French et al. [37]. A standard

analysis for the oxidative stability of lipids in foods is the measurement of TBARS reported by Tarladgis et al. [16], which measures the oxidation product malondialdehyde. Measurement of TBARS levels showed no differences in lipid oxidation levels between meat from NG and CON-fed animals; therefore, contrary to our assumption, NG does not contribute to keeping lipid peroxidation levels of meat low throughout exposure. Gatellier et al. [46] have shown that grazing and mixed feeding (mixed feed consisting of cereal mixture, silage and cattle- cake of different origins) do not affect lipid oxidation. In this study, TBARS levels increased significantly with increasing duration of cold storage (Fig 3B). Reverte et al. [47] observed that forage-fed beef was more susceptible to oxidation than grain-fed beef.

## Conclusions

The results of our study showed that lambs fed CON had higher pre-slaughter weight, HCW, and fat tail weight than lambs fed NG. The NG lambs spent more time eating, drinking, and standing, but less time resting and rumination than the CON-fed lambs. In addition, plasma lipid, β-hydroxybutyrate, and urea levels were higher, while phosphorus levels were lower in NG -fed lambs than in CON-fed lambs. Lambs fed MC and HC had similar growth performance and longissimus dorsi muscle characteristics. These results indicate that lambs fed MC were more efficient in terms of feed efficiency than lambs fed HC and that the use of MC in the diet of fattening lambs is recommended.

## Supporting information

**S1 Table. All raw data on growth performance, behavior, blood parameters, carcass characteristics and meat quality are available in the excel file.** https://doi.org/10.6084/m9.figshare.21617463.v2.
(XLSX)

## Acknowledgments

The authors acknowledge Isfahan University of Technology (Isfahan, Iran) and the Nomadic Affairs Department for the research services. The staff at Lavark Research Station are also thanked for their animal care services. CERCA from Generalitat de Catalunya is also acknowledged.

## Author Contributions

**Conceptualization:** Imaneh Sadrarhami, Masoud Alikhani, Ebrahim Ghasemi, Amir Hossein Mahdavi, Morteza Hosseini Ghaffari.

**Data curation:** Imaneh Sadrarhami, Morteza Hosseini Ghaffari.

**Formal analysis:** Maria Font-i-Furnols.

**Methodology:** Ebrahim Ghasemi, Nafiseh Soltanizadeh, Maria Font-i-Furnols, Morteza Hosseini Ghaffari.

**Project administration:** Amir Hossein Mahdavi.

**Software:** Nafiseh Soltanizadeh, Morteza Hosseini Ghaffari.

**Supervision:** Masoud Alikhani, Ebrahim Ghasemi, Amir Hossein Mahdavi, Maria Font-i-Furnols, Morteza Hosseini Ghaffari.

**Validation:** Morteza Hosseini Ghaffari.

**Visualization:** Morteza Hosseini Ghaffari.

**Writing – original draft:** Morteza Hosseini Ghaffari.

**Writing – review & editing:** Maria Font-i-Furnols, Morteza Hosseini Ghaffari.

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
