## [Decision Letter · Decision Letter 0]

13 Jan 2022

PONE-D-21-28984Effects of nomadic-grazing versus indoor concentrate feeding systems on growth performance, behavior, blood parameters, and carcass quality of finishing lambsPLOS ONE

Dear Dr. Hosseini Ghaffari,

Thank you for submitting your manuscript to PLOS ONE. After careful consideration, we feel that it has merit but does not fully meet PLOS ONE’s publication criteria as it currently stands. Therefore, we invite you to submit a revised version of the manuscript that addresses the points raised during the review process.

ACADEMIC EDITOR:

The manuscript was reviewed by experts and requires major revision before it can be considered for publication. Please take into consideration the points raised by reviewers, especially reviewer#1 when revising your manuscript.

We look forward to receiving your revised manuscript.

Kind regards,

Antonio Humberto Hamad Minervino, Ph.D.

Academic Editor

PLOS ONE

Journal Requirements:

Whilst you may use any professional scientific editing service of your choice, PLOS has partnered with both American Journal Experts (AJE) and Editage to provide discounted services to PLOS authors. Both organizations have experience helping authors meet PLOS guidelines and can provide language editing, translation, manuscript formatting, and figure formatting to ensure your manuscript meets our submission guidelines. To take advantage of our partnership with AJE, visit the AJE website (http://aje.com/go/plos) for a 15% discount off AJE services. To take advantage of our partnership with Editage, visit the Editage website (www.editage.com) and enter referral code PLOSEDIT for a 15% discount off Editage services.  If the PLOS editorial team finds any language issues in text that either AJE or Editage has edited, the service provider will re-edit the text for free.

A clean copy of the edited manuscript (uploaded as the new *manuscript* file)”"

The authors gratefully acknowledge Isfahan University of Technology (Isfahan, Iran) and the Nomadic Affairs Department for the financial support and research services extended to the authors. 

Additional Editor Comments:

The manuscript was reviewed by experts and requires major revision before it can be considered for publication. Please take into consideration the points raised by reviewers, especially reviewer#1 when revising your manuscript.

Reviewers' comments:

Reviewer's Responses to Questions

**Comments to the Author**

1. Is the manuscript technically sound, and do the data support the conclusions?

Reviewer #1: Partly

Reviewer #2: Yes

2. Has the statistical analysis been performed appropriately and rigorously? 

Reviewer #1: Yes

Reviewer #2: Yes

3. Have the authors made all data underlying the findings in their manuscript fully available?

Reviewer #1: Yes

Reviewer #2: Yes

4. Is the manuscript presented in an intelligible fashion and written in standard English?

Reviewer #1: No

Reviewer #2: Yes

5. Review Comments to the Author

Reviewer #1: The conclusion needs to be partly rewritten to reflect the results and to not draw conclusions based on assumptions. Further, the manuscript would benefit from being language edited, both due to grammatical issues as well as incomplete, or wrongfully, sentences.

Reviewer #2: This is an interesting study and the authors have collected a unique dataset using cutting edge methodology. The paper is generally well written and structured with clearly formulated hypothesis of work so I do not have much concern about the design. The aims of the study were to examine the effect of two feeding systems on growth performance, behavior, blood parameters, and carcass quality of finishing lambs. In my opinion the paper has some minor shortcomings. Below I have provided some remarks on the paper.

1- Abstract does not have a suitable structure. In the abstract, after stating the challenges and objectives of the research, the main points of the methodology should be stated and then the results, discussion and general conclusion should be stated.

2- I did not understand how the lambs were slaughtered. Has the HALAL method been performed or not?

3- Authors state that they have allowed 20% of the feed left in the bunk. This method allows the lambs to sort the diet and change their feeding behavior. Are the nutrients in the ort measured to calculate the amount of nutrients consumed?

4- Significant differences in RR and GIT weights between treatments were not correctly interpreted. The mass of these parts is a function of the supply of energy and protein to the animal, which is higher with concentrate diets than with forage diets.

5- Ruminating time is longer with concentrate diets. This result may be a function of the sorting of the given diet or the animal's adaptation mechanisms to concentrate diets. A more appropriate interpretation is needed. It is also recommended that feeding behaviors be stated and interpreted as min to dry matter and fiber intake.

6. PLOS authors have the option to publish the peer review history of their article (what does this mean?). If published, this will include your full peer review and any attached files.

Reviewer #1: No

Reviewer #2: No

---

## [Author Response · Author response to Decision Letter 0]

2 Mar 2022

PLOS ONE 

PONE-D-21-28984

Effects of nomadic-grazing and indoor concentrate feeding systems on growth performance, behavior, blood parameters, and carcass quality of finishing lambs

Journal Requirements:

AU: The manuscript has been revised to meet the style requirements of PLOS ONE.

2. We suggest you thoroughly copyedit your manuscript for language usage, spelling, and grammar. If you do not know anyone who can help you do this, you may wish to consider employing a professional scientific editing service. Whilst you may use any professional scientific editing service of your choice, PLOS has partnered with both American Journal Experts (AJE) and Editage to provide discounted services to PLOS authors. Both organizations have experience helping authors meet PLOS guidelines and can provide language editing, translation, manuscript formatting, and figure formatting to ensure your manuscript meets our submission guidelines. To take advantage of our partnership with AJE, visit the AJE website (http://aje.com/go/plos) for a 15% discount off AJE services. To take advantage of our partnership with Editage, visit the Editage website (www.editage.com) and enter referral code PLOSEDIT for a 15% discount off Editage services. If the PLOS editorial team finds any language issues in text that either AJE or Editage has edited, the service provider will re-edit the text for free. Upon resubmission, please provide the following:

A clean copy of the edited manuscript (uploaded as the new *manuscript* file)”"

AU: We revised the manuscript based on the advice and suggestions provided by Editage's Scientific Editing Service (https://www.editage.com/quality), which specializes in scientific and academic language editing. The revised version was submitted with an editing certificate (Requested under: INQ_IMARH_1) as well.

AU: We revised the manuscript based on the advice and suggestions provided by Editage's Scientific Editing Service (https://www.editage.com/quality), which specializes in scientific and academic language editing. The revised version was submitted with an editing certificate (Requested under: INQ_IMARH_1) as well a copy of your manuscript showing your changes by either highlighting them or using track changes.

The authors gratefully acknowledge Isfahan University of Technology (Isfahan, Iran) and the Nomadic Affairs Department for the financial support and research services extended to the authors. 

AU: We have revised the Acknowledgements Section in order to remove the funding statement. The Funding Statement does not need to be updated.

AU: We prefer not to provide repository information at the acceptance of our data. For this reason, we would like to make changes to our data availability statement. These changes have been described in our cover letter, and our data availability statement has been updated to reflect the new information.

Academic Editor comments:

The manuscript was reviewed by experts and requires major revision before it can be considered for publication. Please take into consideration the points raised by reviewers, especially reviewer#1 when revising your manuscript.

Response to Academic Editor:

Dear Dr. Antonio Humberto Hamad Minervino, 

AU: We appreciate the opportunity to revise and resubmit this manuscript. Thank you for your constructive feedback on our original submission. We greatly appreciate the comments and criticisms. We revised the manuscript based on the advice and suggestions provided by Editage's Scientific Editing Service (https://www.editage.com/quality), which specializes in scientific and academic language editing. The revised version was submitted with an editing certificate (Requested under: INQ_IMARH_1) as well. We have addressed the reviewers' comments and questions in detail below and have made the necessary changes to the text. The changes and additions are highlighted in yellow in the revised manuscript.

Sincerely, 

Morteza H. Ghaffari

However, I would prefer that the manuscript is arranged in the “traditional” way with materials and methods before the results. In my opinion, you need to know what has been done and in what way to be able to interpret the results in a good way. In addition, the manuscript would benefit from being language edited. 

AU: The manuscript was revised based on your guidance on materials and methods before the results and manuscript were provided by Editage's Scientific Editing Service (https://www.editage.com/quality), which specializes in technical and academic language editing.

Main objectives should be described in the abstract. No figures regarding welfare (behaviour) or feed efficiency are mentioned but you draw conclusions about these parameters as well. I would like to see a sentence covering these parameters. The whole abstract would benefit from being rewritten. As it is now you have only used 216 words out of the 300. 

AU: Thank you for your constructive comments. We have revised the Abstract section to reflect the findings according to your suggestions.

pH 24 h which is more interesting than pH at 0 h from an eating quality perspective.

AU: We have revised the Abstract section to reflect the findings according to your suggestions (Lines 24-25).

The second sentence somewhat repeat what is already stated in the first sentence. Reformulate to avoid repetition.

 AU: Revisions were made to prevent repetition.

Also the introduction would benefit from being somewhat rewritten to get the text to flow. For example, the section L 41-58 have a lot of statements but do not really explain why e.g. meat quality is affected. And why is grass feeding discussed just because it is less efficient in growth rate? Here I would like to see pros and cons of both systems. Grazing can be less labour intensive, enhance biodiversity etc.

AU: Thank you for your suggestion. We have revised the Introduction section to reflect the findings according to your suggestions “Considering that feed costs often represent a significant portion of the total variable costs of lamb production, this favorable environment for abundant grass growth results in pasture grass being the most economical feed source [1]. In addition, meat from lambs raised on pasture contains less subcutaneous backfat than meat from lambs raised in pens, which is more attractive to consumers [2]. Therefore, consumers are becoming increasingly interested in sustainable and animal-friendly production and a healthy diet [3].

There has been much discussion about the rearing of grass-fed lambs, as lambs fed concentrates are more efficient than lambs raised on pasture [4]. The improvement of energy supply by concentrates offered to lambs in the barn increases weight gain and eventually carcass yield [5]. In addition, consumption of starch concentrates increases propionate content in the rumen, which stimulates insulin secretion and fat synthesis [6]. The intensity of the production system, as measured by daily growth rate (daily live weight gain), has been shown to have significant effects on meat quality, particularly composition (% lipid) and color [7]. Meat quality can be affected by exercise, especially its color [8]. Previous studies have shown that exercise before slaughter increases the tenderness of the longissimus muscle and drip loss of lambs [9]. Cows that are constantly grazed or kept in an extensive environment have darker muscle color [1].” (Lines 40-55).

I can’t see that Wileman et al. state that consumers show an increasing interest in natural feeding regimes. Please find another source for that statement. With natural feeding, do you mean only grazing or feeding with roughages and cereals etc. without additives?

AU: Thank you for your constructive comments. We have revised the Introduction section to reflect the findings according to your suggestions (Lines 40-42).

You have not studied environmental impact nor profitability, or referred to papers stating this, hence you can’t state that the production systems studied effects these parameters.

AU: Thank you for your constructive comment. Based on your suggestions, we revised the sentence and removed environmental impact and profitability.

You use many abbreviations that haven’t been explained before (they are explained in the materials and methods). I strongly advise you to move the material and methods section and place it before the results, but if you don’t do that you need to explain the abbreviations here.

AU: Your comment has been considered and the text has been modified for abbreviations accordingly. The material and methods section have been moved before the results section.

Personally, I think feed composition is materials and methods rather than results since you have formulated the diets and they are a part of your experimental set up. Hence, I think you should move table 1 to the M&M section. Did you take any samples of the pasture?

AU: Thank you for your constructive comment .Table 1 has been moved to Materials & Methods. Despite taking samples, no analysis has been conducted on the pasture samples.

I miss information about energy content of the diets. Correct unit for DM to g/100 g fresh weight under “Chemical composition” and Add unit for FCR

AU: Thanks for your comment. Metabolizable energy (Mcal/kg DM) of the diets added in Table 1. In response to your question, chemical composition, expressed in terms of dry matter not fresh weight. To determine DM and nutrient composition, representative samples of forages, treatment TMR, and individual refusals were collected immediately before the morning feeding during the 7-d collection period. All samples were frozen immediately at −20 °C until analyses. After thawing, Weekly feed and ort samples were treated at 60 °C in a forced-air oven (Memmert, Schwabach, Germany) for 48 h and then ground to a size less than 1 mm using a Wiley mill (Arthur H. Thomas; Philadelphia, PA, USA). Samples were analyzed in triplicate according to Association of Official Agricultural Chemists (AOAC) [11] for crude protein (CP) by the Kjeldahl method (Kjeltec 1030 Auto Analyzer, Tecator, Höganäs, Sweden; method 955.04), ash (method 942.05), ether extract (method 920.39), and neutral detergent fiber (NDF) using an ANKOM200 Fiber Analyzer (A200 model, Ankom Technology, Macedon, NY) according to the manufacturer's recommended reagents and filter bags (#F57) with pore size of 25 μm. NDF content was analyzed with a heat-stable α-amylase and without sodium sulfite, and was expressed as acid-detergent fiber without considering residual ash. Ash content was measured after combustion in a muffle furnace at 550 °C overnight. The content of non-structural carbohydrates in the feed was calculated on a dry matter basis [10] as follows: 100− (NDF%+CP%+ether extract%+ash%). However, there is no unit associated with FCR.

Present hot carcass weight in kg instead of % and name the figures that you now present as HCW dressing instead, both numbers are interesting.

AU: Thank you for your comment. Hot carcass weight as percentage of carcass weight at slaughter and kg reported on Table 2.

NG lambs also spent more time drinking, could be worth mentioning. Did you measure time moving, walking? Would have been valuable since you state in the introduction that activity affect meat quality.

AU: Thanks for your questions. We were unable to measure walking time in NG lambs due to the difficulty of the measurement.

Figure 1: I assume that the figures presented are accumulated over time? So it is not the changes between the months that is presented? Please clarify.

AU: Thanks for the question. However, drip loss and TBARS level did not differ between the meat of NG and CON-fed animals; instead, as expected, drip loss and lipid oxidation (P<0.01) increased with the duration of storage (Fig 1A–B). 

This figure needs much more explanation to be of value for the reader. The figure (incl. figure text) should be able to explain it self, the reader should not need to read the text to understand. 

AU: In the spirit of your suggestion, the legends of Figures 1 and 2 have been revised to make them self-explanatory.

But you did not have any differences in eating or rumination time between MC and HC? Nor any differences in feed intake? So on what results do you base this statement? Or is this actually what you discuss? In that case, please clarify this part.

AU: We revised the manuscript based on your suggestions and deleted this part from discussion.

The NG lambs did also gain weight and used energy for muscle development, please reformulate.

AU: Thank you for your comment. The EBW and HCW were influenced by finishing system (P<0.05) and were higher in CON-fed lambs than in NG lambs. This difference in HCW is probably due to high energy intake and consequent high growth rate, which from a commercial point of view emphasizes the importance of finishing diets with low indigestible fiber content to achieve maximum live weight gains and ultimately high carcass weights (Lines 376-380).

How do you know that feed intake was higher in NG lambs? They spent more time eating but that could be an indication of lower feed availability and not a higher feed intake per se. Most probably the indoor groups ate more. More drinking time could rather be an indication of a drier feed. How was the weather? Could it also be due to spending more time out in the sun (high temperature)?

AU: Thank you for your constructive comments. We have revised the conclusions section to reflect the findings according to your suggestions. In this study, drinking and standing times were higher (P<0.01), but resting time was lower (P<0.01) in NG lambs than in CON-fed lambs, which was related to finding food for eating and more standing time. It is likely that the indoor groups ate more than the outdoor groups. It is possible that the longer eating time is an indication of lower feed availability rather than higher feed intake per second. In addition, the longer drinking time could be an indication of a drier diet (Lines 434-439).

I would not refer to feedlot as an extensive system.

AU: Thank you for your constructive comment. The feedlot deleted as an extensive system in paragraph.

In what way was L* affected? Please specify.

AU: Thanks for your question. Meat color is influenced by post mortem factors basically involve the refrigerated storage of meat, often termed aging [71]. Protein denaturation during rigor mortis increased free water and juice loss, resulting in changes in light scattering [72]. Therefore, the increase in L* during storage is consistent with the results obtained in chilled beef, suggesting that the difference in L* 24 h post mortem could be partially due to the differences in chilling [73].

But you did not have any differences in chilling?

AU: Thanks for your question. However, drip loss and TBARS level did not differ between the meat of NG and CON-fed animals; instead, as expected, drip loss and lipid oxidation (P<0.01) increased with the duration of storage (Fig 1A–B). 

What is your definition of nomadic pasture? Did you herd the lambs? Please specify that they were housed during night time.

AU: Thank you for your constructive questions. The lambs were randomly assigned to three production systems: NG (n =10), the where animals were housed in loose pens and raised on green nomadic pasture (nomadic livestock breeding on rangelands has been common place as a source of livelihood in different parts of Iran approximately 150 ha) for 14 h per day (0500–1800 h) and they were housed during night time until slaughter. The grazing season lasted 90 days (February 1, 2019 to May 1, 2019). (Lines 82-86).

One pen per group, I assume? How big were the pens? Whit this setup, with all animals in a group, you can’t assess individual feed intakes. It will only be a measure for the whole group and a mean per animal.

AU: Thank you for your questions. The lambs that were handled indoors were housed in a single pen (1.2 m × 1.5 m) (Lines 91-92).

How were the subsamples of feeds and feed residues taken? How samples stored before the subsamples were taken and how were the subsamples stored before analysis?

AU: Thank you for your constructive questions. Samples of forages, subsamples of feeds, and individual refusals were collected immediately before the morning feeding during the 7-d collection period and placed into square nylon. All samples were frozen immediately at −20 °C until analyses. After thawing, weekly feed and ort samples were treated at 60 °C in a forced-air oven (Memmert, Schwabach, Germany) for 48 h and then ground to a size less than 1 mm using a Wiley mill (Arthur H. Thomas; Philadelphia, PA, USA) for each individual lamb (Lines 106-111).

Day 19 – is that 19 days after experimental start for the grazing lambs or is it 19 days after the indoor groups started on their respective treatments? Or were the blood samples not taken on the same day for all groups? Why did you not take blood samples later during the experiment?

AU: In response to your question, blood samples were collected on d 19 after experimental start grazing lambs and indoor groups started by venipuncture of the coccygeal vein (10 mL serum clot activator evacuated tubes) 4 h post-feeding from all lambs as a same day.time (Line 132-134). I did not take blood samples later during the experiment because its many expensive and our financial budget was limited.

Did you measure color on the surface of the LD while still intact on the carcasses? Did you dress that muscle separately from fat? Usually, color is measured after letting the muscle bloom for some time

AU: Thank you for your constructive questions. The meat color of LD muscle at the thirteenth thoracic vertebra was rested 4 h at approximately 7 oC and was then cut into slices, each with a thickness of 2.5 cm (first measurement) and 24 h post mortem (second measurement) (Lines 176-178).

The sentence needs to be rewritten. Did you keep the samples vacuum packed between the drip loss measurements? Or how were the samples stored? And please find another reference for the drip loss. There are several papers dealing with this which are much more suitable than this one about shrimps.

AU: Thank you for your questions. I find another reference for the drip loss:

Honikel KOJMs. Reference methods for the assessment of physical characteristics of meat. Meat Sci, 1998;49(4):447-57. https://doi.org/10.1016/S0309-1740(98)00034-5.

In response to your question, meat samples are cut from the carcass and immediately weighed. A sample weight of approximately 80-100g is recommended. The samples are placed within the container on the supporting mesh and sealed. After a storage period (usually 24 h) at chill temperatures (1 to YC), samples are again weighed. The same samples can be used for further drip loss measurements, e.g. after 1,2,3, 4 months, etc., but in every case the initial weight is used as the reference point. At the time of measurement, samples should be taken immediately from the containers, gently blotted dry and weighed [15] (Lines 190-197).

Response to Reviewer 1 Comments:

The conclusion needs to be partly rewritten to reflect the results and to not draw conclusions based on assumptions. 

AU: Thank you for your constructive comments. We have revised the conclusions section to reflect the findings according to your suggestions. I hope the current draft is clearer and more focused on the research objective associated with the results. 

Further, the manuscript would benefit from being language edited, both due to grammatical issues as well as incomplete, or wrongfully, sentences.

AU: We revised the manuscript based on the advice and suggestions provided by Editage's Scientific Editing Service (https://www.editage.com/quality), which specializes in scientific and academic language editing. The revised version was submitted with an editing certificate (Requested under: INQ_IMARH_1) as well.

Response to Reviewer 2 Comments:

Abstract does not have a suitable structure. In the abstract, after stating the challenges and objectives of the research, the main points of the methodology should be stated and then the results, discussion and general conclusion should be stated.

AU: Thank you for your constructive comment. We have revised the Abstract section according to your suggestions.

I did not understand how the lambs were slaughtered. Has the HALAL method been performed or not?

AU: Yes. Halal method was used. This information has been added in the manuscript.

Authors state that they have allowed 20% of the feed left in the bunk. This method allows the lambs to sort the diet and change their feeding behavior. Are the nutrients in the ort measured to calculate the amount of nutrients consumed?

AU: Yes, they were measured. A subsample of the uneaten feed residues was collected once a week for calculation of nutrient intake.

Significant differences in RR and GIT weights between treatments were not correctly interpreted. The mass of these parts is a function of the supply of energy and protein to the animal, which is higher with concentrate diets than with forage diets.

AU: Your comment has been considered and the text has been modified accordingly as follows: The rumen passage rate of legumes is usually higher than that of grasses because legumes pass through the rumen easily due to particle breakdown [26]. This may be attributed to the fractional rumen passage rate, as grasses may decrease the fractional rumen passage rate and increase the amount of undigested feed reaching the abomasum, which could increase the weight of abomasum tissue [27]. It is important to note that under natural rearing conditions, especially when young ruminants are kept on pasture, most of the nutrients and stimulants for rumen development are obtained from fresh forage. Large particles, fresh forage, and high fiber sources provide physical stimuli that increase rumen motility, muscularization, and rumen volume in calves [28]. Physical stimulation through forage intake increases rumen weight [29]. In fact, RR masses increase due to an increase in RR weight and muscle development [30].

Ruminating time is longer with concentrate diets. This result may be a function of the sorting of the given diet or the animal's adaptation mechanisms to concentrate diets. A more appropriate interpretation is needed. It is also recommended that feeding behaviors be stated and interpreted as min to dry matter and fiber intake.

AU: Thank you for your constructive comment. We fully agree with your comment and the part of this result was not expected. The text has been modified accordingly (Lines 418-434).

---

## [Decision Letter · Decision Letter 1]

20 Apr 2022

PONE-D-21-28984R1Effects of nomadic-grazing and indoor concentrate feeding systems on growth performance, behavior, blood parameters, and carcass quality of finishing lambsPLOS ONE

Dear Dr. Hosseini Ghaffari,

Thank you for submitting your manuscript to PLOS ONE. After careful consideration, we feel that it has merit but does not fully meet PLOS ONE’s publication criteria as it currently stands. Therefore, we invite you to submit a revised version of the manuscript that addresses the points raised during the review process.

ACADEMIC EDITOR: The manuscript was improved but there are still major issues that need to be addressed before it can be considered for publication. Please take into consideration the comments made by the reviewer when revising the manuscript. In addition, I believe that a figure that illustrates the three different feeding systems would improve the manuscript. You can this new figure in the methods section. Pay special attention in the references, as mentioned by the reviewer.

Another suggestion: The table 3 could be transformed into a stacked column graph, providing a better visual representation of your data.

We look forward to receiving your revised manuscript.

Kind regards,

Antonio Humberto Hamad Minervino, Ph.D.

Academic Editor

PLOS ONE

Additional Editor Comments:

The manuscript was improved but there are still major issues that need to be addressed before it can be considered for publication. Please take into consideration the comments made by the reviewer when revising the manuscript. In addition, I believe that a figure that illustrates the three different feeding systems would improve the manuscript. You can this new figure in the methods section. Pay special attention in the references, as mentioned by the reviewer.

Another suggestion: The table 3 could be transformed into a stacked column graph, providing a better visual representation of your data.

Reviewers' comments:

Reviewer's Responses to Questions

**Comments to the Author**

1. If the authors have adequately addressed your comments raised in a previous round of review and you feel that this manuscript is now acceptable for publication, you may indicate that here to bypass the “Comments to the Author” section, enter your conflict of interest statement in the “Confidential to Editor” section, and submit your "Accept" recommendation.

Reviewer #1: (No Response)

2. Is the manuscript technically sound, and do the data support the conclusions?

Reviewer #1: Yes

3. Has the statistical analysis been performed appropriately and rigorously? 

Reviewer #1: Yes

4. Have the authors made all data underlying the findings in their manuscript fully available?

Reviewer #1: Yes

5. Is the manuscript presented in an intelligible fashion and written in standard English?

Reviewer #1: No

6. Review Comments to the Author

Reviewer #1: General comments

The manuscript have improved but there are still some aspects that need to be dealt with before it is suitable for publication. Even though the manuscript has been language edited there still are sentences, or sections, that needs to be rewritten. I have found some of the references that are wrongly used. In addition, you still have a very large number of references and by just reading the title of them I doubt that all of them are correctly used or necessary for this manuscript. Be really careful that you cite the papers correct!

In some cases, it is still confusing if you in the discussion refer to your own study or to other papers. Please, be clear about that.

Abstract

L 22-23 …average age of 90 ± 4 days (mean ± SD)… And the same for 180 days.

L 24 You haven’t explained CON, write out in full first time you mention it.

L 25 What do you mean with “meat values”? If you refer to pH in the meat write instead “higher pH values”.

L 26 I assume you mean “24” and not “4”?

L 35 The fact that MC lambs ate less concentrate is not a result since you decided how much they were offered. Please delete that part.

Introduction

L 44-45 This sentence could be placed first as a statement.

L 49-50 Since you state that consumers prefer less fat, this would be a draw back with concentrate feeding?

L 52 If you refer only to fat content it is better to write that rather than “composition”.

L 53-54 Reference no. 9 did not see any differences neither in WBSF nor in drip loss due to exercise stress. They just mention that others have seen that. Replace the reference or omit the sentence.

L 77-78 Place 90 and 180 outside the parenthesis.

L 83 …the where animals were… Check the sentence and rephrase.

L 83-84 Within parenthesis, I assume you want to say that the pasture you used was 150 ha, the rest of the sentence can be omitted.

L 91-92 “Reared” instead of “handled”. I assume you housed one lamb in each pen but as the sentence is now one can interpret it as you housed all lambs in one single pen, please rephrase.

L 93-94 It is unclear to me what you mean with this sentence. You refer to NRC requirements for dairy cattle and not to a study where the diets are presented.

L 95 The lambs did not refuse feed intake, they left 20% refusals of feed offered. Rephrase the sentence.

L 97 Did you only take samples of the orts to analyse nutrient composition? Lambs are good at sorting the feed so by only analysing the orts you will not get a picture of what the lambs actually have eaten. Or do you mean that you calculated the feed intake? Did you feed forages and concentrates separately or as TMR? Please clarify.

L 107 Square nylon? Please specify what you mean.

L 121 Just some “left overs” from earlier versions, I assume.

Table 1 Yes, I understand that the chemical composition is presented as g/100 g DM but DM itself is not g/100 g DM (then it would be 100% in your table), rather g/100 g fresh weight. All other components are in g/100 g DM, as you correctly have stated.

L 132-133 Please, rephrase the sentence.

L 153 Either delete NG or rephrase.

L 153-154 Move this sentence to “Animals and experimental design”

L 158-159 You slaughtered all animals the same day (180) and not by a target weight, please rephrase.

L 165-166 Review the sentence.

L 174 On line 168 you state that you took a sample from the right half of the carcasses for meat quality measurements but here you state that you measured on the left half. Which is correct?

L 176-178 How you conducted the colour measurements is still confusing to me. Did you cut out the LD muscle right after slaughter and sliced it? Did you measure on the same piece of meat all the time or on different slices? Or did you measure on several slices at each time? Why did you do it immediately after slaughter? The most common is to do it after chilling and cutting the carcass. Did you proceed with the other meat quality measurements on the same sample? Cutting out the LD at such an early stage will effect e.g tenderness.

L 184 The sample was then weighed. Should this sentence be there?

L 190-197 This part is directly taken from the book “Handbook of Muscle Foods Analysis”. It is not ok to take a text verbatim from another source!

L 205-212 WHC is a property of raw meat that describes the ability to bind water. It can be measured in different ways, e.g. as drip loss. When you heat the meat it is rather cooking loss you get. I can’t see that the method you describe is presented in the paper of Graham Trout. Why centrifuge at 10,000 g, when this is usually done at ordinary gravity (1 g) when it comes to drip loss?

L 277-278 You have no values of feed intake or FCR for NG. Reformulate to levels of concentrate instead of feeding indoors.

L 281-282 Loin thickness did only differ between NG and CON, not between MC and HC. Please specify.

L 284, Table 2 RR is a part of the GIT, as you now have defined it in M&M, and should be presented as such also here.

Table 3 It is enough of you present the result in whole minutes. Since you assumed that the behaviour continued for 5 min you don’t have that precision to actually measure in seconds.

L 322 …indicated that chemical composition of the meat did not differ…

L 330 Delete “…in the CON fed lamb meat than that in NG lamb meat.”

Table 5 Cooking loss, is it really in g/kg? WHC, what does the figures really mean? As said before, WHC is usually measured as drip loss so the latter would have been more proper to present in the table.

Discussion

L 371-373 These two sentences say almost the same (ADG= weight gain).

L 384 Again, RR is a part of GIT.

L 400 I assume you mean low fat cover for NG lambs? However, in your study you could not see a difference in back fat cover among the treatments. Or do you refer to another study? This is also valid for L 401-403, you own or someone else’s study?

L 414 Rephrase “…the consume energy and energy cost…”

L 418 “Ruminating” instead of “rumination”.

L 426-427 This sentence do not contribute to the text and can be omitted.

L 437 There was only one outdoor group.

L 437-438 I think it is quite obvious that longer eating time is due to lower feed availability. Review if you really need the long reasoning above to come to that conclusion. And It should not be “…higher feed intake per second.” But “…per se”.

L 441-450 But is longer eating time always a bad thing, is the welfare worse? Of course, if the animals are starving but their natural behaviour is to seek for feed a large part of the day. I miss a discussion about “optimal” eating time, which automatically does not lead to a lower welfare due to longer eating time.

L 480 Meat quality

L 497 “…in the mixed group”, you need to explain what this means.

L 504 Rephrase “…, that aging improved…”.

L 505-510 But did you have differences in chilling between the treatments? If so, this is confounding and should have been stated in the M&M. If you did not have different chilling, why do you think this could be the reason?

L 516 Pre-slaughter

L 518 Is 75 really the right reference? Earlier you referred to another paper when writing about WHC. And once again, WHC is a property of the raw meat and drip loss is used to describe it. Cooking loss is cooking loss and nothing else.

Conclusion

L 549 To state that the NG lambs had lower welfare just due to longer eating time is doubtful. Then you need to put that in relation to what a “normal” eating time is. I think you should delete this statement.

L 551 You haven’t discussed profitability at all in the discussion. I agree that they probably are since they ate less concentrate but performed as well as HC. However, to state this I think you should mentioned this in the discussion as well.

7. PLOS authors have the option to publish the peer review history of their article (what does this mean?). If published, this will include your full peer review and any attached files.

Reviewer #1: No

---

## [Author Response · Author response to Decision Letter 1]

18 May 2022

ACADEMIC EDITOR: The manuscript was improved but there are still major issues that need to be addressed before it can be considered for publication. Please take into consideration the comments made by the reviewer when revising the manuscript. In addition, I believe that a figure that illustrates the three different feeding systems would improve the manuscript. You can this new figure in the methods section. Pay special attention in the references, as mentioned by the reviewer.

AU: Thank you for your comments. The manuscript has been revised based on your comments. The section on materials and method has been updated with illustrations showing different feeding systems. In addition, some of the references have been deleted or changed.

Another suggestion: The table 3 could be transformed into a stacked column graph, providing a better visual representation of your data.

AU: Thank you for your suggestion. We transferred table 3 to column graph to better visual representation of our data.

General comments

The manuscript have improved but there are still some aspects that need to be dealt with before it is suitable for publication. Even though the manuscript has been language edited there still are sentences, or sections, that needs to be rewritten. I have found some of the references that are wrongly used. In addition, you still have a very large number of references and by just reading the title of them I doubt that all of them are correctly used or necessary for this manuscript. Be really careful that you cite the papers correct! 

In some cases, it is still confusing if you in the discussion refer to your own study or to other papers. Please, be clear about that. 

AU: Thank you for your comments. The sentences have been reworded as much as possible. The number of references have been reduced. We have reviewed all references and deleted some; others have been corrected or revised based on citations in the paper. In writing, we tried to eliminate doubts and change or delete them when they were not needed in the sentences.

Abstract

…average age of 90 ± 4 days (mean ± SD) and the same for 180 days. 

AU: Thank you for your comments. According to your suggestions, was corrected.

You haven’t explained CON, write out in full first time you mention it. 

 AU: Thank you for your constructive comments. We have revised this section according to your suggestions.

What do you mean with “meat values”? If you refer to pH in the meat write instead “higher pH values”.

AU: Thank you for your suggestion, we have revised the sentence.

I assume you mean “24” and not “4”?

AU: Thank you for your comment. Yes, the 24 is right and it was corrected.

The f act that MC lambs ate less concentrate is not a result since you decided how much they were offered. Please delete that part. 

AU: Thank you for your constructive comments. We have removed that part and corrected.

Introduction

This sentence could be placed first as a statement.

AU: Your comment has been considered and the sentence have been moved first as a statement (Lines 39-40). 

Since you state that consumers prefer less fat, this would be a draw back with concentrate feeding?

AU: Thank you for your comment. Yes, you are right. Meat from lambs raised on pasture contains less subcutaneous backfat than meat from lambs raised in pens, which is more attractive to consumers [2]. Diaz et al [2] reported that in sheepfold lamb dorsal fat thickness, kidney knob and cannel fat, and percent leg fat were higher in lambs raised in pens than in lambs raised on pasture. Lambs with higher weight had more fat. Considering that feed costs often represent a significant portion of the total variable costs of lamb production, this favorable environment for abundant grass growth results in pasture grass being the most economical feed source. In addition, pasture grass has a high content of natural antioxidants and therefore provides animals with better protection against lipid oxidation [3]. According to Santé-Lhoutellier et al [4], the oxidative stability of lamb is related to its diet. Pasture feeding offers some advantages over concentrate feeding in terms of lipid oxidation and, to a lesser extent, protein oxidation [4].

There has been much discussion about raising grass-fed lambs because lambs fed concentrates are more efficient than lambs raised on pasture [5]. By giving concentrates to lambs in the barn, it is possible to improve weight gain, carcass yield, and ultimately the profitability of the production system [6]. Color of meat is also an important factor in consumer selection. In particular, exercise can influence color and flavor [7]. It has been shown that cows that are constantly on pasture or kept in an extensive environment have a darker muscle color [8] (Lines 40-56).

If you refer only to fat content it is better to write that rather than “composition”.

AU: Thank you for your comment and your comment was used.

Reference no. 9 did not see any differences neither in WBSF nor in drip loss due to exercise stress. They just mention that others have seen that. Replace the reference or omit the sentence.

AU: Thank you for your detailed examination. The reference no. 9 was deleted. 

Place 90 and 180 outside the parenthesis.

AU: Your comment has been considered and we place 90 and 180 outside the parenthesis. 

…the where animals were… Check the sentence and rephrase.

AU: Thank you very much for your comment. The sentence has been rewritten.

Within parenthesis, I assume you want to say that the pasture you used was 150 ha, the rest of the sentence can be omitted.

AU: Thanks for your explanation and the sentence was shortened. 

“Reared” instead of “handled”. I assume you housed one lamb in each pen but as the sentence is now one can interpret it as you housed all lambs in one single pen, please rephrase.

AU: Thank you for your constructive comment and we used “Reared” instead of “handled”.

It is unclear to me what you mean with this sentence. You refer to NRC requirements for dairy cattle and not to a study where the diets are presented.

AU: Thank you very much for your comment. The reference was corrected and we used the NRC (2007) requirements for lamb (reference no. 9). 

The lambs did not refuse feed intake, they left 20% refusals of feed offered. Rephrase the sentence.

AU: Your comment has been considered and rephrase the sentence based on your comment (Lines 99-100). 

Did you only take samples of the orts to analyse nutrient composition? Lambs are good at sorting the feed so by only analysing the orts you will not get a picture of what the lambs actually have eaten. Or do you mean that you calculated the feed intake? Did you feed forages and concentrates separately or as TMR? Please clarify.

AU: Thanks for your questions. The sentence was rewritten. Each morning, the entire backyard was removed and the feed consumption was then measured. Representative samples were taken from each stall for further analysis (Lines 100-102). We feed forages and concentrates as TMR.

Square nylon? Please specify what you mean.

AU: Thank you for your attention. I mean nylon bag stock and it has been corrected.

Just some “left overs” from earlier versions, I assume.

AU: Thank you for your careful look and yours was right. The sentence was deleted. 

Yes, I understand that the chemical composition is presented as g/100 g DM but DM itself is not g/100 g DM (then it would be 100% in your table), rather g/100 g fresh weight. All other components are in g/100 g DM, as you correctly have stated.

AU: Thank you for your comment. Yes, your comment is right and it was corrected and Dry matter is % of fresh fed weight. It has been clarified in Table 1.

Please, rephrase the sentence.

AU: Thank you for your comment. The sentence was rephrased. 

Either delete NG or rephrase.

AU: Thank you for your comment and NG has been deleted. 

Move this sentence to “Animals and experimental design”.

AU: Thank you for your suggestion and this sentence has been moved to “Animals and experimental design”.

You slaughtered all animals the same day (180) and not by a target weight, please rephrase.

AU: Thank you for your constructive comment. The sentence has been rephrased.

Review the sentence.

AU: Thank you for your comment. The sentence was review and deleted because it not necessary.

On line 168 you state that you took a sample from the right half of the carcasses for meat quality measurements but here you state that you measured on the left half. Which is correct?

AU: Thank you for your careful review and your question. Right half is correct and the sentence has been corrected. 

How you conducted the color measurements is still confusing to me. Did you cut out the LD muscle right after slaughter and sliced it? Did you measure on the same piece of meat all the time or on different slices? Or did you measure on several slices at each time? Why did you do it immediately after slaughter? The most common is to do it after chilling and cutting the carcass. Did you proceed with the other meat quality measurements on the same sample? Cutting out the LD at such an early stage will effect e.g tenderness. 

AU: Thank you for your questions. I'm sorry to confusing you. 

Answer to question 1: Yes, LD muscles are cut out and sliced immediately after slaughter.

Answer to question 2: The different pieces of meat (muscle LD) we used for the measurement. However, the color of the meat was obtained from the thirteenth thoracic vertebra immediately after slaughter and at 24 hours post mortem in the same piece.

Answer to question 3: The color of meat depends on the concentration of a haeminic pigment, myoglobin, and its oxidation state (Renerre, 1990). Color measurements were taken immediately after slaughter and 24 hours postmortem to determine the effects of storage time (Sen et al., 2011) and color stability.

Answer to question 4: The other measurements were made on the longissimus dorsi muscle, but I used the same sample to measure color 0- and 24-hours postmortem. 

Answer to question 5: Longissimus dorsi muscle was excised from the right half carcass and sampled for the following meat quality determinations immediately after slaughter (Lines 167-169). The color of a meat cut (2.5 cm-thick) from the thirteenth thoracic vertebra, was determined at 0 h post mortem (immediately after slaughter) and 24 h post mortem in the same section (Lines 175-177). 

Response to Question 6: We agree with the reviewer's position. Removal of the LD at an early stage may result in tenderness, and we can take that into consideration. However, the objective was to compare the 3 treatments and, the effect of cutting at an early stage was identical for all of them.

 The sample was then weighed. Should this sentence be there?

AU: Thanks for your comment. This sentence has been deleted.

This part is directly taken from the book “Handbook of Muscle Foods Analysis”. It is not ok to take a text verbatim from another source!

AU: Thank you for your comment. An edited sentence replaced the previous one (Lines 188-193). 

WHC is a property of raw meat that describes the ability to bind water. It can be measured in different ways, e.g. as drip loss. When you heat the meat it is rather cooking loss you get. I can’t see that the method you describe is presented in the paper of Graham Trout. Why centrifuge at 10,000 g, when this is usually done at ordinary gravity (1 g) when it comes to drip loss?

AU: Thank you for your attractive question. According to your comments, WHC has been removed from manuscript. 

Results

You have no values of feed intake or FCR for NG. Reformulate to levels of concentrate instead of feeding indoors.

AU: Thank you for your suggestion and this sentence reformulate to levels of concentrate.

Loin thickness did only differ between NG and CON, not between MC and HC. Please specify.

AU: Thank you for your comment and your comment was done (Lines 272-273).

RR is a part of the GIT, as you now have defined it in M&M, and should be presented as such also here.

AU: Thank you for your comment. We decided to enter the performance parameters related to slaughter in the new table after blood parameters and behavior, and your suggestion done (Lines 302-304).

It is enough of you present the result in whole minutes. Since you assumed that the behaviour continued for 5 min you don’t have that precision to actually measure in seconds.

AU: Thank you for your constructive comment. Data were arranged in minutes and presented in bar plots (fig. 2) instead of as a table.

…indicated that chemical composition of the meat did not differ…

AU: Thank you for your comments, it has been changed accordingly.

Delete “…in the CON fed lamb meat than that in NG lamb meat.”

AU: Thank you for your offer and your offer has been made.

Cooking loss, is it really in g/kg? WHC, what does the figures really mean? As said before, WHC is usually measured as drip loss so the latter would have been more proper to present in the table.

AU: Thank you for your questions. The unit of cooking loss (%) was corrected. As I mentioned before, according to your comments, WHC has been removed from manuscript. 

Discussion

These two sentences say almost the same (ADG= weight gain).

AU: Thank you for your comment and one sentence has been deleted because they were same. 

Again, RR is a part of GIT.

AU: Thank you for your comment and it has been corrected.

I assume you mean low fat cover for NG lambs? However, in your study you could not see a difference in back fat cover among the treatments. Or do you refer to another study? This is also valid for L 401-403; you own or someone else’s study?

AU: Thank you for your comment. Back fat cover was not different among the treatments. I refer to another study and it has been corrected.

Rephrase “…the consume energy and energy cost…”

AU: Your comment has been considered and the sentence has been deleted.

“Ruminating” instead of “rumination”.

AU: Thanks for the comment and it has been changed.

This sentence does not contribute to the text and can be omitted. 

AU: Your suggestions has been done and this sentence has been deleted.

There was only one outdoor group.

AU: Thank you and in response to your comment, there was only one outdoor group and the sentence was corrected. 

I think it is quite obvious that longer eating time is due to lower feed availability. Review if you really need the long reasoning above to come to that conclusion. 

AU: Your comment has been considered and the sentence has been deleted. 

But is longer eating time always a bad thing, is the welfare worse? Of course, if the animals are starving but their natural behavior is to seek for feed a large part of the day. I miss a discussion 2 “optimal” eating time, which automatically does not lead to a lower welfare due to longer eating time.

AU: Thank you for your comment. Eating time is related to fattening system and feed availability. Based on feed availability, based on feed availability, Salem et al [23] found that the time spent grazing herbaceous vegetation actually corresponds to the time of searching vegetation residues above ground. Feed restriction may induce animals to increase their eating rate to maximize intake during periods of feed availability (Lines 362-365).

AU: Thank you for your suggestion and it has been changed. 

“…in the mixed group”, you need to explain what this means.

AU: Thank you for your comment and was corrected.

Rephrase “…, that aging improved…”.

AU: Thank you for your comments. I deleted “…, that aging improved…” at the end of the sentence because it was not necessary and corrected this sentence (Lines 450-456).

But did you have differences in chilling between the treatments? If so, this is confounding and should have been stated in the M&M. If you did not have different chilling, why do you think this could be the reason?

AU: Thank you for your question. No, we haven’t differences in chilling between treatments. This sentence was deleted and used other reference.

Pre-slaughter

AU: Thank you and has been changed accordingly to your comment.

Is 75 really the right reference? Earlier you referred to another paper when writing about WHC. And once again, WHC is a property of the raw meat and drip loss is used to describe it. Cooking loss is cooking loss and nothing else.

AU: Thank you for your constructive comment and we have deleted this parameter to reflect the findings to your suggestions. 

Conclusion

To state that the NG lambs had lower welfare just due to longer eating time is doubtful. Then you need to put that in relation to what a “normal” eating time is. I think you should delete this statement.

AU: Thank you for your comment and this statement has been deleted. 

You haven’t discussed profitability at all in the discussion. I agree that they probably are since they ate less concentrate but performed as well as HC. However, to state this I think you should mentioned this in the discussion as well. 

AU: I appreciate your comment. This was mentioned in this discussion. We found that lambs fed MC and HC had similar performance and carcass quality. It is debatable whether MC is an attractive option for finishing lambs or for maintaining lamb growth when concentrate prices are relatively high compared to lamb meat. On a commercial level, this could be beneficial in situations where concentrate prices are relatively high compared to lamb meat (Lines 424-428).

---

## [Decision Letter · Decision Letter 2]

10 Jun 2022

PONE-D-21-28984R2Effects of nomadic-grazing and indoor concentrate feeding systems on growth performance, behavior, blood parameters, carcass characteristics, and meat quality of finishing lambsPLOS ONE

Dear Dr. Hosseini Ghaffari,

Thank you for submitting your manuscript to PLOS ONE. After careful consideration, we feel that it has merit but does not fully meet PLOS ONE’s publication criteria as it currently stands. Therefore, we invite you to submit a revised version of the manuscript that addresses the points raised during the review process.

ACADEMIC EDITOR: The manuscript can be accepted for publication after minor correction pointed by reviewer. 

We look forward to receiving your revised manuscript.

Kind regards,

Antonio Humberto Hamad Minervino, Ph.D.

Academic Editor

PLOS ONE

Journal Requirements:

Additional Editor Comments (if provided):

The manuscript can be accepted for publication after minor correction pointed by reviewer.

Reviewers' comments:

Reviewer's Responses to Questions

**Comments to the Author**

1. If the authors have adequately addressed your comments raised in a previous round of review and you feel that this manuscript is now acceptable for publication, you may indicate that here to bypass the “Comments to the Author” section, enter your conflict of interest statement in the “Confidential to Editor” section, and submit your "Accept" recommendation.

Reviewer #1: All comments have been addressed

2. Is the manuscript technically sound, and do the data support the conclusions?

Reviewer #1: Yes

3. Has the statistical analysis been performed appropriately and rigorously? 

Reviewer #1: Yes

4. Have the authors made all data underlying the findings in their manuscript fully available?

Reviewer #1: Yes

5. Is the manuscript presented in an intelligible fashion and written in standard English?

Reviewer #1: Yes

6. Review Comments to the Author

Reviewer #1: The manuscript has improved substantially. After some small changes I find it suitable for publication.

Abstract

L 22-23 50:50% and 30:70%, make clear that it is on DM basis

L 28 You expected the CON-fed lambs to have better carcass yield so delete “interestingly”.

L 35 You haven’t explained MC and HC, you can preferably do so in connection to 50:50 and 30:70 on lined 22-23.

Materials and methods

L 101 It is unclear to me what you mean with “backyard”.

Results

L 303 Weight is already included in EBW and HCW so you can delete “weight”.

L 303 and Table 4. I still think that RR should be included in GIT, not presented by itself since it is a part of the whole gastrointestinal tract. If you want to present it specifically you need to clearly state what you include in GIT (e.g. GIT excl RR).

Discussion

L 403 84%

L 413 Think about how you want to present GIT and RR.

L 439 Glucose is one part in lowering pH after slaughter, but also glycogen has an important role. I think you can omit the part “… because of glucose level was not different.”

7. PLOS authors have the option to publish the peer review history of their article (what does this mean?). If published, this will include your full peer review and any attached files.

Reviewer #1: No

---

## [Author Response · Author response to Decision Letter 2]

29 Jun 2022

Dear Dr. Antonio Humberto Hamad Minervino, 

AU: We appreciate the opportunity to revise and resubmit this manuscript. Thank you for your constructive feedback on our original submission. We greatly appreciate the comments and criticisms. We revised the manuscript based on the advice and have addressed the reviewers' comments. 

Sincerely, 

Morteza H. Ghaffari

General comments

L 22-23 50:50% and 30:70%, make clear that it is on DM basis

AU: Thank you for your comment and your comment was used (Lines 22-23).

L 28 You expected the CON-fed lambs to have better carcass yield so delete “interestingly”.

AU: Thank you for your comment and “interestingly” has been deleted.

L 35 You haven’t explained MC and HC, you can preferably do so in connection to 50:50 and 30:70 on lined 22-23.

AU: Thank you for your suggestion and this part has been moved to lined 22-23.

Materials and methods

L 101 It is unclear to me what you mean with “backyard”.

AU:Thank you for your comments. “backyard” has been deleted and I added “during each feeding, orts were collected and weighed before feed was given to the animals” (Lines 101-102).

Results

L 303 Weight is already included in EBW and HCW so you can delete “weight”.

AU: Thank you for your careful review and “weight” has been deleted (Lines 304).

L 303 and Table 4. I still think that RR should be included in GIT, not presented by itself since it is a part of the whole gastrointestinal tract. If you want to present it specifically you need to clearly state what you include in GIT (e.g. GIT excl RR).

AU: Thank you for your suggestion and RR parameter has been removed from manuscript.

Discussion

L 403 84%

AU: Thank you for your comment and it has been corrected (Line 403).

L 413 Think about how you want to present GIT and RR.

AU: Thank you for your suggestion. As I said before, I want to deleted RR parameter.

L 439 Glucose is one part in lowering pH after slaughter, but also glycogen has an important role. I think you can omit the part “… because of glucose level was not different.”

AU: Thank you for your comment and it has been deleted (Line 438).

Journal Requirements:

AU: All references have been checked to ensure that they are correct and complete, and to avoid citing work that has been retracted.

---

## [Decision Letter · Decision Letter 3]

22 Aug 2022

PONE-D-21-28984R3Effects of nomadic-grazing and indoor concentrate feeding systems on growth performance, behavior, blood parameters, carcass characteristics, and meat quality of finishing lambsPLOS ONE

Dear Dr. Hosseini Ghaffari,

Thank you for submitting your manuscript to PLOS ONE. After careful consideration, we feel that it has merit but does not fully meet PLOS ONE’s publication criteria as it currently stands. Therefore, we invite you to submit a revised version of the manuscript that addresses the points raised during the review process.

ACADEMIC EDITOR: Dear authors, unfortunately we have some delay from the original reviewer to responde the invitation to revise this manuscript again, which resulted in the invitation of a new reviewer. Subsequently, the original reviewer accepted to revise the manuscript and now I got a split decision (original reviewer: accept; new reviewer: reject). Thus, I invite the authors to read carefully the comments made by the reviewers and revise the manuscript accordingly.

We look forward to receiving your revised manuscript.

Kind regards,

Antonio Humberto Hamad Minervino, Ph.D.

Academic Editor

PLOS ONE

Additional Editor Comments (if provided):

Dear authors, unfortunately we have some delay from the original reviewer to responde the invitation to revise this manuscript again, which resulted in the invitation of a new reviewer. Subsequently, the original reviewer accepted to revise the manuscript and now I got a split decision (original reviewer: accept; new reviewer: reject). Thus, I invite the authors to read carefully the comments made by the reviewers and revise the manuscript accordingly.

Reviewers' comments:

Reviewer's Responses to Questions

**Comments to the Author**

1. If the authors have adequately addressed your comments raised in a previous round of review and you feel that this manuscript is now acceptable for publication, you may indicate that here to bypass the “Comments to the Author” section, enter your conflict of interest statement in the “Confidential to Editor” section, and submit your "Accept" recommendation.

Reviewer #1: All comments have been addressed

Reviewer #3: (No Response)

2. Is the manuscript technically sound, and do the data support the conclusions?

Reviewer #1: Yes

Reviewer #3: No

3. Has the statistical analysis been performed appropriately and rigorously? 

Reviewer #1: Yes

Reviewer #3: No

4. Have the authors made all data underlying the findings in their manuscript fully available?

Reviewer #1: Yes

Reviewer #3: Yes

5. Is the manuscript presented in an intelligible fashion and written in standard English?

Reviewer #1: Yes

Reviewer #3: No

6. Review Comments to the Author

Reviewer #1: (No Response)

Reviewer #3: This is interesting research, but the experimental design shows several weaknesses. The differences between the production systems would be due in large part to differences in nutrient intake, but feed intake in grazing animals was not recorded. In addition, there is only one fedlot per experimental treatment. Additionally, the manuscript has several aspects to improve, which are described in the specific comments.

SPECIFIC COMMENTS

Title. Please remove the nomadic term. In my humble opinion, it involves herd movement around a large geographic area. In this study a rotational grazing system was used probably in a small area (less than 1 km2).

Abstract. Please remove paragragh related to profitability and sustainability. Neither production costs nor environmental impact were evaluated.

Introduction. There are many studies comparing grazing and intensive production systems based on TMR diets, and even the ratio of concentrate to forage on lamb performance and meat quality. The interest of this research has not been clearly described. What does this research contribute with respect to previous knowledge?

Material & methods:

• What breed was used?

• Were all animals male lambs?

• Please indicate grazing area.

• Was forage cut before mixing it with concentrate feeds?

• Were orts daily collected or every week?

• What was the total area of the trough? Were all the animals able to access the feeder at the same time? Do the authors think there might be differences between lambs in diet selection?

• Line 127. What does it mean (the NDF was expressed as ADF)?

• Line 212. Strange did not determine TBARs.

• Table 1. Ash content values seems to be too low [if you remove mineral content of additives (calcium carbonate, oxid manganese, salt, premix), an ash content value less than 0.5% can be estimated for the feed ingredients (98.5% of the total diet); it means that the mean value of organic matter content of the feed ingredients (alfalfa, straw, barley, …) would be greater than 99%].

• Was behaviour assessed on only one day?; Do the authors think that one day of behavioral recording is representative of the whole experimental period?

• Was LD dissected inmediately after slaughter? Was not carcass refrigerated during 24 hours?

• Meat samples for texture and TBARS analysis were frozen, but for drip loss determination they were storaged under refrigerated conditions for 4 months (Lines 193-196) and reused. Is this interpretation correct? Does it make any sense to keep lamb meat for 4 months refrigerated at 4 º C?

• Please modify description of meat chemical analysis. For instance: i) line 216. What does it mean? Was not the meat freeze-dried and then ground before the chemical analysis?; ii) Please remove intramuscular, as CP, fat and ash are intramuscular components in this case.

• How were loin thickness and fat thickness measured?

• Statistical analysis. Meat colour at 0 and 24 h postmortem was measured on the same slice (Line 180). Therefore, a repeated measurement analysis should have been performed. In contrast, TBARS at 0,1, 2 and 3 months were determined on different slices, so a repeated measurement analysis would not be strictly necessary.

Results

• Table 2. i) Please provide correct units: feed intake was g DM/day); FCR was g of DM/g ADG; ii) it was not possible to compare the FCR of MC and HC because there is only one value per treatment; iii) mean ADG value should not be very different from that estimated as the difference between the initial and final mean LBW divided by 90 day (for NG: (41.80-28.32/90 = 0.15 kg/day, but the mean value showed in table was 0.19).

• Table 4. i) Liver and GIT weights are not carcass characteristics; ii) mean values should be checked. Slaughter weight estimated by dividing HCW by dressing percentage is much higher than the final LBW showed in table 2 (NG: 41.8 vs 43.4; MC: 46.5 vs 49.4; HC: 47.7 vs 50.6).

Discussion. This section needs to be rewritten. It should be focused on results avoiding:

• Speculations. For example, feed intake, and rate of passage and particle size reduction were not recorded. NG lambs spent less time ruminating and it could be due to a lower NDF intake which could be due to a lower dry matter intake but also to a diet with lower NDF content (diet selection). Authors do not know the reason of the different behavior. These results should be interpreted with caution as they were collected for only 1 day and the experimental period lasted 90 days.

• Mistakes. Starch intake does not increase rumination time (Line 371).

• Confusing paragraphs. For instance, lines 366-368. What does mean that feed restriction may induce animals to increase their eating rate?; lines 368-369. What does mean to distinguish between rumen function and animal welfare? Line 422. What does “masculinization of rumen” mean?

• The use of inappropriate references. Line 435. Foti et al. evaluated the effect in suckling lambs, no in fattening lambs.

• Inadequate interpretation of the results. Lines 478-479. What are the data suggesting a high lipid oxidation in NG lambs?

Conclusions must be rewritten based on results. i) What parameters evaluating efficience were determined?; ii) what parameter were used to asses carcass quality? Only HCW?

7. PLOS authors have the option to publish the peer review history of their article (what does this mean?). If published, this will include your full peer review and any attached files.

Reviewer #1: No

Reviewer #3: No

---

## [Author Response · Author response to Decision Letter 3]

13 Sep 2022

SPECIFIC COMMENTS

Title: 

•Please remove the nomadic term. In my humble opinion, it involves herd movement around a large geographic area. In this study a rotational grazing system was used probably in a small area (less than 1 km2).

AU: Thank you for your comments and your explanation. In this study, nomadic grazing had a regular seasonal pattern of movement so if you allow the title of the article to be nomadic grazing.

Abstract:

•Please remove paragraph related to profitability and sustainability. Neither production costs nor environmental impact were evaluated.

AU: Your comment has been considered and remove paragraph related to profitability and sustainability.

Introduction: 

•There are many studies comparing grazing and intensive production systems based on TMR diets, and even the ratio of concentrate to forage on lamb performance and meat quality. The interest of this research has not been clearly described. What does this research contribute with respect to previous knowledge?

AU: Thank you for your constructive comments. The requested paragraph has been added “Therefore, the appropriate production system and the weight of lambs before slaughter are of great importance in lamb production to obtain high quality lamb carcasses. However, the interest of this research was that feeding management plays an important role in feeding behavior, which leads to changes in plasma parameters and meat quality that are not well understood. This experiment will help provide more data and a better understanding of these relationships to support strategic feeding management to increase performance and improve carcass characteristics.”(Lines 57-62).

Material & methods:

• What breed was used?

AU: Thank you very much for your question and the study was carried in central Iran (Isfahan province, Iran) during the grazing season lasted 90 days (February 1, 2019 to May 1, 2019) using Turki-Ghashghaei breed sheep (Lines 75-76).

• Were all animal’s male lambs?

AU: Thanks for your question. Yes, all were male lambs.

• Please indicate grazing area.

Thank you for your attention and rotational grazing (RG, approximately 150 ha). The RG lambs were reared in Aghdash (31°36′09″ N 51°32′32″ E) in Vardasht Rural District of Semirom City (Isfahan Province, Iran) (Lines 86-88).

• Was forage cut before mixing it with concentrate feeds?

AU: Thank you for your careful review and your question. Yes, forage was cut before mixing it with concentrate feeds. The desired paragraph was added. As part of the harvesting process (Golchin Trasher Hay Co., Isfahan, Iran), alfalfa was chopped to a length of 30 mm by using a harvester with a screen size regulator. In the harvesting process, wheat straw was chopped to a theoretical cut length of 10 mm by using a harvester (Golchin Trasher Hay Co., Isfahan, Iran) (Lines 102-105).

• Were orts daily collected or every week?

AU: Thank you for your attractive question. During each feeding, orts were collected daily and weighed before feed was given to the animals, and the feed consumption was then measured (Lines 109-110).

 • What was the total area of the trough?

AU: Thank you for your question. The desired paragraph was added. The total area of the trough was 30 cm (Lines 107-108).

• Were all the animals able to access the feeder at the same time? 

AU: Thank you for your question. Yes, the desired paragraph was added. All the animals able to access the individual trough (30 cm) at the same time (Lines 107-108).

• Do the authors think there might be differences between lambs in diet selection?

AU: Thank you for your questions. Yes, we think there might be differences between lambs in diet selection and sorting index from CON-fed lambs were measured (data not shown). 

• Line 127. What does it mean (the NDF was expressed as ADF)?

AU: Thank you for your question. It was removed from paragraph. 

• Line 212. Strange did not determine TBARs.

AU: Thank you for your comment and changed by another reference ( Tarladgis BG, Watts BG, Younathan MT, Dugan JL. A distillation method for the quantitative determination of malonaldehyde in rancid foods. J. Am. Oil Chem. 1960; 37:44– 48).

• Table 1. Ash content values seems to be too low [if you remove mineral content of additives (calcium carbonate, oxid manganese, salt, premix), an ash content value less than 0.5% can be estimated for the feed ingredients (98.5% of the total diet); it means that the mean value of organic matter content of the feed ingredients (alfalfa, straw, barley, …) would be greater than 99%].

AU: The authors are grateful for your careful consideration. I apologize for the typo and the ash content has been corrected (Table 1). 

• Was behaviour assessed on only one day? Do the authors think that one day of behavioral recording is representative of the whole experimental period?

AU: Thank you and to answer your question, it was very difficult to measure animal behavior under grazing conditions because the animals were grazing and their behavior was recorded every 5 minutes for 24 hours. In other sources, animal behavior was measured once for 24 hours during the whole period (Hamidreza Mirzaei-Alamouti et al., 2021).

• Was LD dissected immediately after slaughter? Was not carcass refrigerated during 24 hours?

AU: Thank you for your questions. Yes, the longissimus dorsi muscle was taken from the right half of the carcass and sampled for the following meat quality determinations immediately after slaughter. The color of a piece of meat (2.5 cm thick) from the thirteenth thoracic vertebra was determined 0 h post mortem (immediately after slaughter) and 24 h post mortem in the same section (lines 183-185). The color of meat depends on the concentration of a haeminic pigment, myoglobin, and its oxidation state (Renerre, 1990). Color measurements were taken immediately after slaughter and 24 hours postmortem to determine the effects of storage time (Sen et al., 2011). For this reason, we had to sample the other meat quality parameters at the same time and immediately after slaughter to ensure uniformity of the meat samples.

•Meat samples for texture and TBARS analysis were frozen, but for drip loss determination they were storage under refrigerated conditions for 4 months (Lines 193-196) and reused. Is this interpretation correct? Does it make any sense to keep lamb meat for 4 months refrigerated at 4 º C?

AU: We appreciate your constructive question. I'm sorry to confusing you. For drip losses and TBARS determination a sample of 80 and 100 g was placed from a plastic nylon bag stock in a refrigerated room (4ºC) for 24 h and other samples placed from freeze rome (-16 ºC) for 1, 2, 3 and 4 months (Lines 196-198).

• Please modify description of meat chemical analysis.

AU: Thank you for your constructive comment and the description was modify Meat quality and texture characteristics.

•For instance: line 216. What does it mean? Was not the meat freeze-dried and then ground before the chemical analysis?

AU: We highly appreciate important question. Meat chemical analysis samples were vacuum packed, frozen and stored at -20 °C until further analysis (AOAC, 2005). To measure fat, after dehydrating the meat using a freeze dryer were freeze‐dried, ground, and then it was measured (Lines 223-225).

•Please remove intramuscular, as CP, fat and ash are intramuscular components in this case.

AU: Thank you for your suggestion and intramuscular removed.

• How were loin thickness and fat thickness measured?

AU: Thank you for your question. An ultrasound scanner (Ultrascan 50, Linear 3.5 MHz Scan, Alliance Medical Inc) was used to measure the thickness of backfat and loin fat in live lambs between the third and fourth lumbar vertebrae (halfway between the last rib and the hip bone). (Lines 114-117).

• Statistical analysis. Meat colour at 0 and 24 h postmortem was measured on the same slice. Therefore, a repeated measurement analysis should have been performed. In contrast, TBARS at 0,1, 2 and 3 months were determined on different slices, so a repeated measurement analysis would not be strictly necessary.

AU: Thank you for your constructive comment and your comment done. Data with repeated measures (color of LD muscle) were analyzed with PROC MIXED from SAS (SAS 9.1; SAS Institute Inc; Cary, NC, USA) when variables were measured over time (Line 253).

Results:

• Table 2. Please provide correct units: feed intake was g DM/day); FCR was g of DM/g ADG 

AU: Thank you very much for your comment. Feed intake was corrected as a kg DM/day. FCR was corrected as a kg of DM/kg ADG (Table 2).

•It was not possible to compare the FCR of MC and HC because there is only one value per treatment;

AU: Thank you for your comment. There is not only one value per treatments. I measured FCR for all lambs (CON- fed lambs: 20 numbers of data) and analysis by SAS (SAS 9.1; SAS Institute Inc; Cary, NC, USA). 

•Mean ADG value should not be very different from that estimated as the difference between the initial and final mean LBW divided by 90 day (for NG: (41.80-28.32/90 = 0.15 kg/day, but the mean value showed in table was 0.19).

AU: Thank you for your questions. NO, ADG for all lamb calculated and put it 30 numbers of data for ADG to SAS (SAS 9.1; SAS Institute Inc; Cary, NC, USA) and data were extracted.

• Table 4. i) Liver and GIT weights are not carcass characteristics;

AU: The authors are very grateful for your comment. I removed Liver and GIT weights from carcass characteristics and I added Non-carcass parts for this parameters (Table 4).

•Mean values should be checked. Slaughter weight estimated by dividing HCW by dressing percentage is much higher than the final LBW showed in table 2 (NG: 41.8 vs 43.4; MC: 46.5 vs 49.4; HC: 47.7 vs 50.6).

AU: Thank you for your comment. Slaughter weight was not estimated by dividing HCW by dressing percentage. I changed this parameters as a Pre-slaughter BW, kg because were measured before being transferred to the slaughter route due to stress before slaughter (Table 2).

Discussion: 

•This section needs to be rewritten. It should be focused on results avoiding

AU: Thank you for your constructive comments and I deleted some paragraph that data were not recorded.

•Speculations. For example, feed intake, and rate of passage and particle size reduction were not recorded. NG lambs spent less time ruminating and it could be due to a lower NDF intake which could be due to a lower dry matter intake but also to a diet with lower NDF content (diet selection). Authors do not know the reason of the different behavior. These results should be interpreted with caution as they were collected for only 1 day and the experimental period lasted 90 days.

AU: Your comment has been considered and the sentence have been deleted.

•Mistakes. Starch intake does not increase rumination time (Line 371).

AU: We appreciate your comment. This paragraph has been deleted.

• Confusing paragraphs. For instance, lines 366-368. What does mean that feed restriction may induce animals to increase their eating rate?

AU: Thank you for your comment and paragraph has been deleted. 

• Lines 368-369. What does mean to distinguish between rumen function and animal welfare? 

AU: The authors appreciate the delicate view of the reviewer. Rumination has been considered as a key component of digestion and intake by ruminants and as well as an important role in animal welfare (Krawczel and Grant, 2009). The natural behaviors that are most important to the health, welfare, and productivity of cows are resting, feeding, and rumination.

•Line 422. What does “masculinization of rumen” mean?

AU: Thank you for your question. The development of the rumen epithelium and muscularization are differently affected by the nature of the diet. Feed physical structure likely has the greatest influence on development of rumen masculinization and volume. Stimulation of rumen motility is governed by the same factors, particle size and effective fiber (Van Soest 1994). Feeding concentrate feeds in early life stimulates the development of the epithelium, while forages with large particle size or high fiber sources appear to be the primary stimulators of rumen masculinization and volume (Zitnan et al., 1998).

• The use of inappropriate references. Line 435. Foti et al. evaluated the effect in suckling lambs, no in fattening lambs.

AU: Thank you for your careful look. Foti et al. Was removed

• Inadequate interpretation of the results. Lines 478-479. What are the data suggesting a high lipid oxidation in NG lambs?

AU: Thanks for your comment. This sentence was deleted. 

Conclusions:

• Must be rewritten based on results. i) What parameters evaluating efficiency were determined?

AU: Thank you for your question and this paragraph was removed. 

•what parameter were used to assec carcass quality? Only HCW?

AU: Thank you for your question. The sentence was corrected based on your comment. The results of our study showed that lambs fed CON were higher pre-slaughter weight, HCW, and fat tail weight than lambs fed NG (Lines 479-480).

---

## [Editor Report · Decision Letter 4]

22 Nov 2022

Effects of nomatic grazing system and indoor concentrate feeding systems on growth performance, behavior, blood parameters, carcass characteristics, and meat quality of finishing lambs

PONE-D-21-28984R4

Dear Dr. Hosseini Ghaffari,

We’re pleased to inform you that your manuscript has been judged scientifically suitable for publication and will be formally accepted for publication once it meets all outstanding technical requirements.

Kind regards,

Antonio Humberto Hamad Minervino, Ph.D.

Academic Editor

PLOS ONE

Additional Editor Comments (optional):

Dear authors,

I am glad to inform that your manuscript was satisfactorily revised and now it can be accepted for publication at PLoS One.

I have one minor suggestion: please reduce the tittle, is too big.
---

## [Editor Report · Acceptance letter]

29 Nov 2022

PONE-D-21-28984R4 

Effects of nomadic grazing system and indoor concentrate feeding systems on performance, behavior, blood parameters, and meat quality of finishing lambs 

Dear Dr. Hosseini Ghaffari:

I'm pleased to inform you that your manuscript has been deemed suitable for publication in PLOS ONE. Congratulations! Your manuscript is now with our production department. 

Kind regards, 

on behalf of

Dr. Antonio Humberto Hamad Minervino 

Academic Editor

PLOS ONE